**DOI: 10.1038/ncomms13953**　　**OPEN**

# Mate choice in fruit flies is rational and adaptive

Devin Arbuthnott[1,2], Tatyana Y. Fedina[3], Scott D. Pletcher[3] & Daniel E.L. Promislow[1,4]

According to rational choice theory, beneficial preferences should lead individuals to sort available options into linear, transitive hierarchies, although the extent to which non-human animals behave rationally is unclear. Here we demonstrate that mate choice in the fruit fly *Drosophila melanogaster* results in the linear sorting of a set of diverse isogenic female lines, unambiguously demonstrating the hallmark of rational behaviour, transitivity. These rational choices are associated with direct benefits, enabling males to maximize offspring production. Furthermore, we demonstrate that female behaviours and cues act redundantly in mate detection and assessment, as rational mate choice largely persists when visual or chemical sensory modalities are impaired, but not when both are impaired. Transitivity in mate choice demonstrates that the quality of potential mates varies significantly among genotypes, and that males and females behave in such a way as to facilitate adaptive mate choice.

[1] Department of Pathology, University of Washington, 1959 NE Pacific Street, Box 357705, Seattle, Washington 98195, USA. [2] Department of Zoology, University of British Columbia, 4200-6270 University Boulevard, Vancouver, British Columbia, Canada V6T 1Z4. [3] Department of Molecular and Integrative Physiology, and Geriatrics Center, University of Michigan, 109 Zina Pitcher Place, Ann Arbor, Michigan 48109, USA. [4] Department of Biology, University of Washington, Seattle, Washington 98195, USA. Correspondence and requests for materials should be addressed to D.A. (email: darbuthnott@gmail.com).

Do humans and other animals make rational decisions? This question has been the subject of much debate in economic and psychological research, as its answer has significant implications for group behaviour, well-being and economic patterns[1,2]. Rational choices made by individuals are based on maximizing benefits and will result in observable and predictable behavioural outcomes among groups. The major hallmark of rational choice is transitivity (if A > B and B > C, then A > C)[3,4]. Transitivity comes about because the relative benefits of each choice should remain fixed, which leads to a linear rank order of options, and choosers should favour the highest ranking option in any scenario.

Humans often behave irrationally, showing intransitive choice[5] and altering their choices based on the presence of irrelevant alternatives[6], which leads to disadvantageous choices in a number of contexts. The extent to which non-human animals act rationally has received less attention[2] and several studies have found inconsistent results[7–9]. Recent theoretical work from an evolutionary perspective[10,11] has also challenged the assumption that adaptive decision rules necessarily produce transitive choices. From this challenge, researchers have introduced the concept of 'ecological rationality'[11], where choice rules evolve to maximize fitness in the environment to which a population has adapted. As environments and the availability of alternative options can change rapidly, adaptive decision rules can produce non-transitive choices, although such choices are still rational as they maximize fitness. However, for much of the animal kingdom, we do not know the extent to which choices are transitive, nor whether or when non-transitive choices might be rational. To fully understand behavioural evolution of any organism, it is critical to know whether individuals of any species make transitive and rational choices, as transitive choices reflect variation both in quality among alternatives and in the chooser's ability to detect and evaluate such variation. Each of these factors, as well as the fitness consequences of decisions, influence the potential for quality, the signals of that quality, and preferences for these signals to respond to selection.

Individuals make rational choices to maximize benefits and, in an evolutionary sense, the choice of who to mate with can have drastic impacts on fitness. An individual that chooses a mate with good genes or abundant resources can greatly increase the number, survival, and reproductive potential of his or her offspring[12]. Despite the evolutionary importance of mate choice, studies of animal rationality have primarily focused on foraging behaviour[7,8] or social dominance hierarchies[13]. Transitive mate choice is very rarely discussed or considered in sexual selection research and, to our knowledge, transitivity of mate choice in animals has only been tested twice[14,15] and has shown inconsistent results.

Of the few tests that have been done on mate choice rationality[14–16], each presented females with potential mates that differed in traits affecting just one sensory modality. Yet, sexual displays are often complex, signalling to several different sensory modalities and differing in many ways within modalities[17], and individuals of both sexes are faced with making mating decisions based on all of the information available rather than on single traits in isolation. In fact, rational choice theory assumes that individuals have complete information about their choices[1], and that they may need to process all information available to choose the option with the greatest cumulative benefits if two options are very similar along several axes of assessment[3]. As mating displays are complex and the relationships between separate display traits are unclear, it does not necessarily follow that linear preferences for individual sexual traits will lead to mate choice transitivity.

In fact, recent theoretical work[10,11] has suggested that preferences that violate transitivity may be adaptive in fluctuating environments. Given the complications discussed here, one cannot assume that observed linear preferences for single traits, which are typically the focus in sexual selection literature, will lead to transitive choices among potential mates when those mates express multiple sexual signals simultaneously.

The most informative way to assess mate choice transitivity is to measure mating preferences when individuals choose among alternative potential mates that vary in all relevant axes simultaneously and to experimentally replicate this whole-individual variation. Such a framework would not only allow us to evaluate the transitivity of mate choices but would also allow us to assess the nature of the information conveyed by separate sexual traits. Sexual cues and displays are made up of several separate traits employing a number of sensory modalities[17] and individuals have access to the information signalled by each of these traits simultaneously. However, the nature of the 'complete' information that is required from these multiple sexual traits for rational mate choice is unclear. If separate traits signal orthogonal components of a potential mate's fitness[18], individuals might need information from most or all of each of a potential mate's sexual signals to make rational choices. Alternatively, if the multiple traits act as redundant cues[19], individuals could make rational choices based on very few traits.

In this study, we characterize the mate choice of fruit flies when males are given the choice between females from alternative inbred lines. As each female line is inbred, all individuals within each line are genetically identical, which allows us to observe mate choice for specific genotypes rather than particular traits in isolation, and to test for the hallmarks of rational decision-making in a repeatable way. Furthermore, we are able to measure how mate choice is affected by female behavioural and non-behavioural traits, and the relationship among such traits. Specifically, we are able to determine whether separate female traits act redundantly in enabling males to detect and assess female quality or whether separate female traits signal orthogonal components of fitness. We are also able to measure the fitness consequences of the observed mate choices.

We find that mate choice is transitive, such that female genotypes are consistently and linearly ranked during mate choice, and that such rankings are repeatable across male populations. When males are given any pairwise choice of females, the chosen female tends to be the one that produces the most offspring, qualifying these transitive mate choices as rational. Such rational choices persist when males are unable to see or smell/taste females, but not if both senses are impaired. We therefore conclude that the transitive mate choices of fruit flies are beneficial and based on redundant female cues that facilitate the detection and assessment of potential mates.

## Results

**Mate choice transitivity.** To assess the extent to which fruit flies act rationally when choosing mates, we carried out mate choice trials, where individual males were given a choice between two females from a set of ten unique female genotypes from inbred lines of the *Drosophila* Genetic Reference Panel (DGRP[20]). Although males are the sex given the choice in potential mates in these trials, both male and female behaviours probably contribute to the resulting mate choice. The female lines were generated by collecting wild fruit flies and subjecting them to 20 generations of inbreeding[20]. Therefore, even though these lines are inbred, their unique genotypes encompass natural genetic variation from a wild population. We exposed individual virgin

males to all possible pairwise combinations of females from the ten lines, to assess the preferences and transitivity of mate choice in this species. Although transitive choice is typically considered a property of individuals making a series of choices, we only tested each male once, to control for potential changes in mating behaviour through time and sexual experience. We therefore were measuring population-level preferences and transitivity rather than individual-level transitivity. When discussing mate choice in our results, we are referring to this group-level preference.

When we exposed outbred wild-type Canton-S males to all 45 pairwise combinations of females from ten DGRP lines, we found that the resulting mate choice was significantly transitive (proportion of transitive triads, $t_{tri} = 0.95$, $P = 0.0003$), indicating that these ten inbred female lines represent a linear hierarchy with respect to male mating bias (Fig. 1). A replicate study with Canton-S males showed consistent patterns of transitivity ($t_{tri} = 0.95$, $P = 0.0003$) and mating bias among each of the 45 pairwise combinations of the 10 female genotypes in the 2 replicates was significantly correlated (Pearson's correlation, $r = 0.48$, $P = 0.0009$). Furthermore, male mating biases in an unrelated wild-type strain, Oregon-R, also showed significant transitivity ($t_{tri} = 0.99$, $P < 0.00001$) that was significantly correlated with the mating biases of Canton-S males (Fig. 2a). Thus, when outbred males are given the choice between alternate female genotypes, mate choice is consistent, transitive and repeatable, both within and among populations, suggesting that genetically distinct males respond similarly to repeatable female properties. Although population-level, or 'vote-based' measurements of choice can mask individual-level transitivity if different individuals have different transitive preferences[2,21,22], the striking similarity in the choices of these two unrelated populations of males suggests that our group-level assessment of choice reflects individual preferences and transitivity.

**Redundancy of sensory modalities in mate choice.** Having demonstrated repeatable transitivity in mate choice, we next characterized the nature of the information used to make such rational choices. To determine whether mate choice is transitive when males have limited information and whether female traits signal redundant or orthogonal components of sexual fitness, we measured mate choice when male sensory perception was impaired, manipulating sight, smell/taste or both. We manipulated male sight by either carrying out mate choice trials with blind $ninaB^{-/-}$ mutant males or carrying out trials in the dark. We manipulated male smell/taste by carrying out mate choice trials with $ppk23^{-/-}$ mutant males, which are deficient in detecting specific female cuticular hydrocarbons (CHCs), and impaired both sight and smell/taste by carrying out trials with $ppk23^{-/-}$ mutant males in the dark. When a single sensory modality was impaired, mate choice was still significantly transitive, whether males were deprived of access to visual signals (Canton-S males in the dark: proportion of transitive triads, $t_{tri} = 1$, $P = 0.00001$; $ninaB^{-/-}$ blind mutants: $t_{tri} = 1$, $P = 0.00001$) or access to specific female chemical cues ($ppk23^{-/-}$ mutants: $t_{tri} = 0.97$, $P = 0.00006$). The mating biases of each of these groups of males were significantly correlated with the biases of wild-type males under standard conditions (Fig. 2a). Therefore, mate choice is similar and transitive even when males have limited information.

However, when both visual and chemical perceptions were impaired ($ppk23^{-/-}$ flies in the dark), males' mating rate dropped drastically (Fig. 2b), to the point that we could not assess their mate preferences. These data show that males use sight and smell/taste to detect females, leading to transitive mate choices, but do not need both to do so. Therefore, mate choice of female genotypes is consistent even when males have imperfect information from just one sense or the other. This result indicates that separate female sexual cues convey

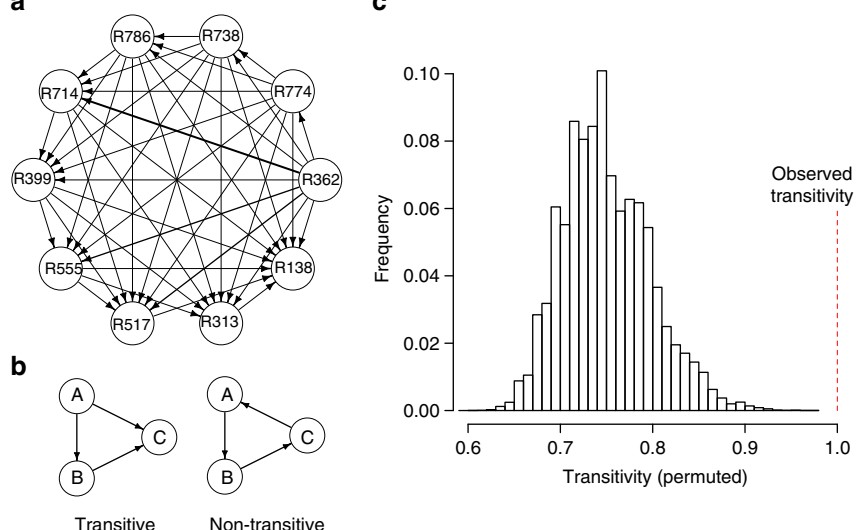

**Figure 1 | Male *D. melanogaster* display transitive mate choice.** (**a**) Mate choice network for Canton-S males, combining two independent replicates. More attractive lines (a line that received more than half of the matings in any pairwise competition) point to less attractive lines. The width of a line is proportional to the mating skew for each pairwise interaction. Absent arrows (for example, R517 versus R313) denote ties between two lines. Line number (for example, R362) corresponds to DGRP ID. (**b**) Examples of transitive and non-transitive triads. The overall transitivity of a network, $t_{tri}$, is calculated as the proportion of transitive triads in the full network. (**c**) Transitivity scores from 100,000 permutations of mate choice data, used to calculate the significance of our mate choice network. On average, we expect that 75% of relationships would be transitive at random, as 6/8 possible relationships within a triad are transitive. Dashed line represents the observed transitivity score for Canton-S males, which is significantly greater than transitivity scores expected to be generated at random. The data used to generate panels (**a**) and (**c**) are the combined results of the separate mate choice assays using Canton-S males, while the statistics reported in the text were calculated for each assay separately.

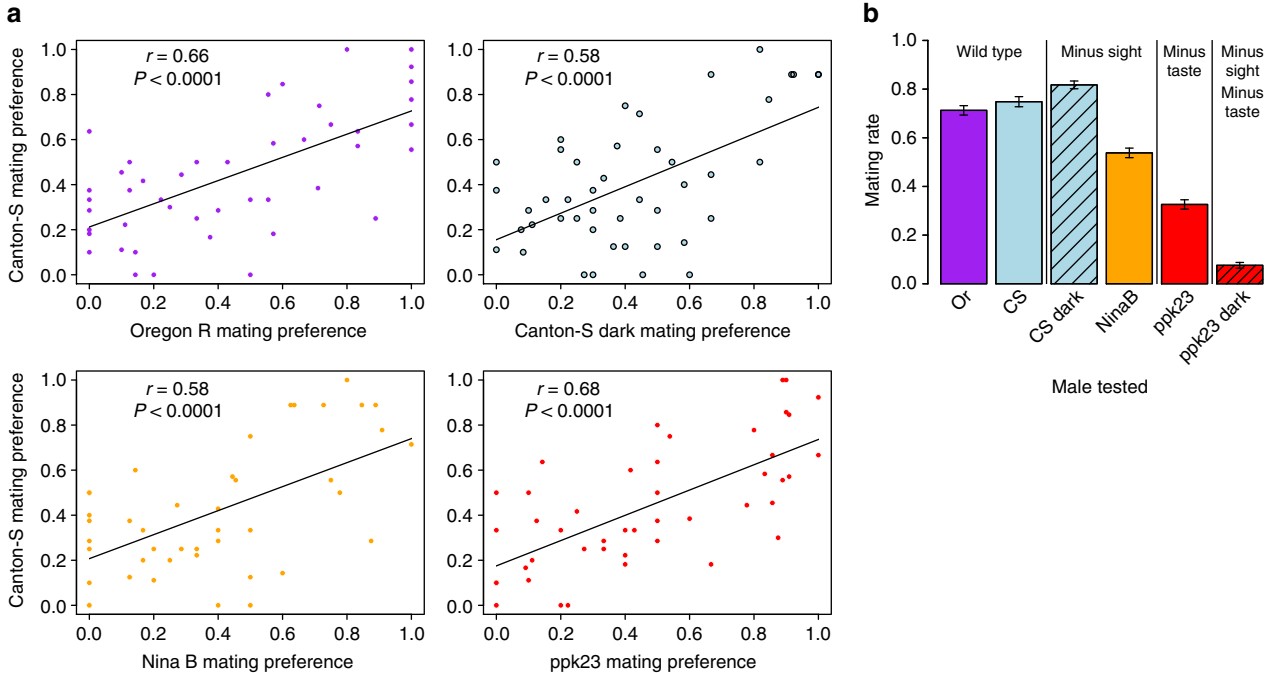

**Figure 2 | Males with impaired signal detection make similar mate choices.** (**a**) Correlation of the proportion of the first female line mated for all 45 pairwise combinations of 10 DGRP female lines between Canton-S males and Oregon-R (wild type), Canton-S males in the dark (blind), $ninaB^{-/-}$ mutant males (blind) and $ppk23^{-/-}$ mutant males (deficient CHC detection). Regression lines represent geometric mean regression. The significance of Pearson's correlation coefficients were calculated via 10,000 permutations of the data (see Methods for details). (**b**) Mating rates for all males tested (± s.e.m.). Bar colour represents genetic background (purple = Oregon-R, blue = Canton-S, orange = $ninaB^{-/-}$, red = $ppk23^{-/-}$), whereas black hashmarks signify that mating trials were conducted in the dark.

information in a correlated manner. This built-in redundancy ensures transitive mate choices even in environments in which some signals are blocked.

**Female traits and mate choice.** To understand how transitive mate choices persist with imperfect information, we measured the effects of three traits that have previously been demonstrated to play a role in *Drosophila* male mate choice trials: female body size[23], CHCs[24] (Supplementary Fig. 1), and female receptivity[25]. Male *D. melanogaster* have been shown to display preferences for larger females, presumably due the positive correlation between size and fecundity. Although we saw significant differences in female body size among the ten inbred lines (analysis of variance, $F_{9,67} = 16.6$, $P = 6.1 \times 10^{-14}$), female size did not influence the mate choice of any of the males tested (Supplementary Fig. 2a). Past studies in this species that demonstrated male preference for larger females presented males with extreme dichotomous choices between large and small females, whereas female size in our experiments was more reflective of natural variation.

We examined correlations between each measured CHC (Supplementary Fig. 1) and attractiveness ranking among the ten inbred lines based on the choice of Canton-S males. Two peaks, CHC9 and CHC10, corresponding to 9,13-PD and 7,11-PD, respectively, showed substantial negative correlations between relative abundance and female line ranking ($r < -0.5$ for both), so we examined these CHCs further. There were strong negative correlations between the relative abundance of one or both of these hydrocarbons and male mating bias for all males tested, except the $ppk23^{-/-}$ mutant, consistent with its deficiency in CHC detection (Supplementary Fig. 2b,c), suggesting that high levels of these CHCs act as repellent traits.

We measured female receptivity to mating via the time from courtship to mating, with shorter times signifying more receptive females. There were significant differences among the lines in female mating receptivity (likelihood ratio test (LRT), $\chi^2_1 = 71.1$, $P = 3.0 \times 10^{-12}$), although the DGRP lines' receptivity to wild type versus $ppk23^{-/-}$ males was uncorrelated (Pearson's correlation, $r = 0.3$, $P = 0.43$), meaning that male lines pursue females differently and/or females alter their receptivity depending on the males with which they interact. Female receptivity was significantly correlated with the mate choice of wild-type males (Pearson's correlation with significance tested via permutations, $r = 0.81$, $P = 0.00001$), although this relationship was weak for the choices of $ppk23^{-/-}$ males ($r = 0.3$, $P = 0.04$; Supplementary Fig. 2d). For all males tested, the greater the difference in receptivity between female lines, the greater the preference that males showed towards the more receptive female.

**Male versus female agency in mate choice.** The correlation between female receptivity and mating biases in male choice trials raises the question of whether transitive sorting of mates is the product of male behaviour, female behaviour, or both. To clarify the relative roles of male and female behaviour during our male choice trials, we carried out additional analyses and experiments. First, we exposed individual males (Canton-S and $ppk23^{-/-}$) to single females of each genotype and measured the lag time between male introduction and courtship, which is a proxy for male motivation to mate these females. We found a weak but significant negative correlation between time to courtship and male mate choice in two-female choice experiments for each male tested (Fig. 3a). This result suggests that, on average, the faster the males court females in a no-choice

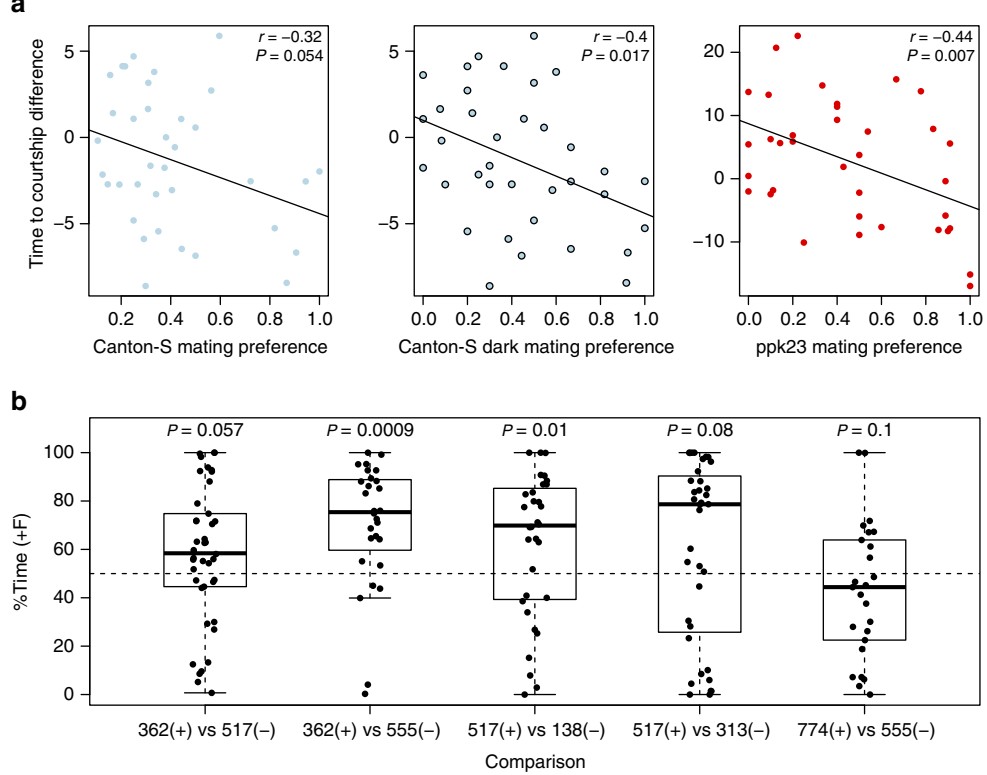

**Figure 3 | Evidence for male choice among female genotypes.** (**a**) Relationship between time to courtship in single-male single-female pairings and the mate choice of males in two-choice trials. Significance of Pearson's correlation coefficients were calculated via 10,000 randomizations of the data (see methods for details). (**b**) The proportion of time males spend with the more attractive female (as determined by previous trials with active females) when males interact with two decapitated females. The specific female line choices are given along the x axis, with the attractive ( + ) and unattractive ( − ) female denoted for each comparison. P-values are from Wilcoxon signed-rank tests against the null hypothesis of 50% time with the more attractive female. Box boundaries represent s.e.m. and whiskers show 80th and 20th percentiles of data.

scenario, the more likely they are to choose that female's genotype in a choice experiment. Although latency to courtship is thought to be indicative of male choice, it can also be influenced by female receptivity and motivation to mate. We therefore attempted to statistically correct for female receptivity in the mating biases of CS males (see Methods). When we accounted for variation explained by female receptivity, we found that the resulting corrected mate choices were significantly correlated with those from the uncorrected data (Pearson's correlation with significance assessed via permutations, $r = 0.59$, $P = 0.0001$) and were still significantly transitive, although to a lesser degree than the uncorrected data (proportion of transitive triads, $t_{tri} = 0.88$, $P = 0.011$).

Although these analyses suggest that female receptivity plays a crucial role in producing the observed mate choices, by statistically correcting for the variance explained by female receptivity, we are potentially ignoring variance explained by non-behavioural traits that are correlated with female receptivity. To further clarify the role of female receptivity in these mate choices, we also carried out additional assays to observe male preferences in the absence of female behaviour. Specifically, we measured the time males spent in proximity to two decapitated (and therefore non-responding) females[26,27], with genotype combinations chosen from a subset of the previously used female line combinations. We found significant or marginally nonsignificant male preference for the more attractive (as determined in active female assays) female genotype in four out of five tested female comparisons (Fig. 3b) and one comparison yielded no significant preference for either

female. A Fisher's combined probability test revealed a significant trend for the five inactive female comparisons to agree with the results of the active female assays ($\chi^2_{10} = 34.2$, $P = 0.0002$). This finding supports the interpretation that male choice plays a significant role in many of our active female mate choice assays. However, the fact that in one out of five tested female combinations, male choice between decapitated females did not recapitulate preference for intact females suggests that in some instances female receptivity could drive the outcome of mate choice. Therefore, it appears that the observed transitive mate choice is a product of both male and female behaviours and their interactions.

**Fitness benefits of mate choice**. Transitivity in choice is thought to arise because individuals make rational choices to their benefit. To assess the benefit of transitive mate choice to males, we measured offspring production in the DGRP female lines when mated to wild-type (Canton-S) or $ppk23^{-/-}$ males. There were significant differences among the female lines with respect to offspring production (LRT, $\chi^2_1 = 105.05$, $P = 3.2 \times 10^{-19}$) and these differences were consistent whether females were mated to Canton-S or $ppk23^{-/-}$ males (Pearson's correlation, $r = 0.87$, $P = 0.003$). Although female offspring production was not significantly correlated with any of the measured female traits associated with mating biases in male choice trials (Fig. 4a), several of these relationships were heavily influenced by one highly productive but unattractive line (line 517, Fig. 1). We identified this line as a significant outlier for the relationships

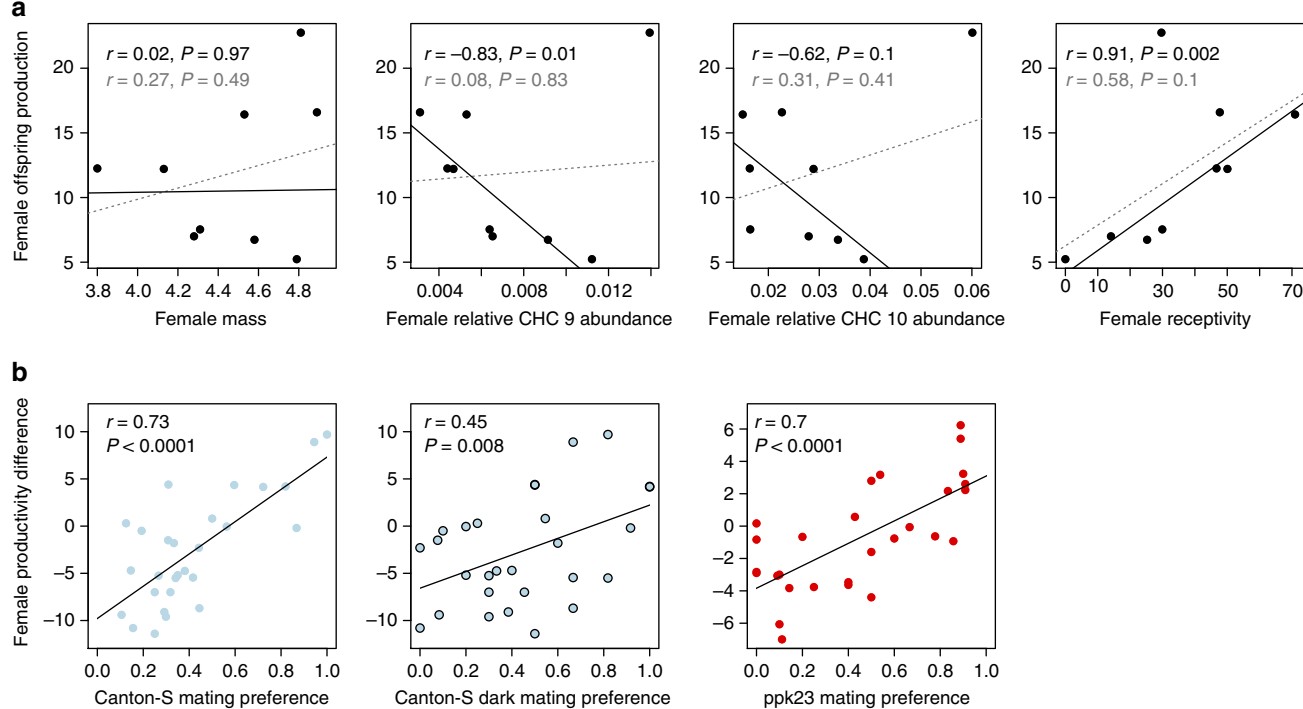

**Figure 4 | Male mating preferences coincide with female offspring production.** (**a**) Relationships between female offspring production and mass, relative abundance of CHC 9, relative abundance of CHC 10 and female receptivity. Black lines and text show correlations between female traits with data from an outlier isogenic line removed, whereas grey lines and text show correlations with all data. Reported correlation coefficients and *P*-values are from Pearson's correlation. (**b**) Correlations between the proportions of the first female line mated and pairwise differences in female offspring production when an outlier line was removed. Significance was tested via permutation tests to avoid pseudoreplication. Regression lines for all plots represent geometric mean regression.

between productivity and both CHC 9 abundance and female receptivity (Bonferonni outlier analysis based on a linear model; $P = 0.013$ and $P = 0.008$, respectively). When this outlier was removed, there were significant correlations between female productivity and both CHC 9 abundance and female receptivity (Fig. 4a). More importantly, when this outlier line was removed, mate choice was significantly correlated with female productivity for all males tested (Fig. 4b). Therefore, fruit flies' transitive mate choice matched our prediction that males tend to mate the most productive female when exposed to any combination of females, qualifying these mate choices as rational. The causes of the outlier line's phenotypic mismatch between productivity and attractiveness are unclear at present, but should be investigated further. The mechanisms and frequency of the genetic decoupling of female attractiveness and quality could have important implications for the benefits of the observed mate choices, as well as the evolutionary dynamics of mate choice. We also evaluated the relationship between mate choice after correction for female receptivity and found that these adjusted preferences did not correlate with female offspring production (Pearson's correlation with significance tested via permutations, $r = 0.004$, $P = 0.98$). Therefore, once again, it appears that female behaviour plays an integral role in mediating the link between preferences and offspring production. However, our overall results suggest that by biasing matings towards specific female genotypes and cues, males maximize their own fitness, resulting in rational mate choices.

## Discussion

We have demonstrated that mate choice in fruit flies shows the major hallmark of rational decision-making, transitivity, and

that this transitivity is repeatable across males from several unrelated populations. We used males as the choosers in our assays to demonstrate adaptive rational mate choice because, despite accumulating evidence for the presence of male mate choice[28,29] and its importance in evolution[23], the assumption that males act as eager but indiscriminate mating partners remains pervasive. When we carried out assays that removed female response behaviours from sexual interactions, we found agreement with male choice between two female lines whether females were responding (unmanipulated) or non-responding (decapitated) in most, but not all, instances. Therefore, although male behaviour appears to play an important role in the observed mate choices, female behaviour also appears to significantly affect mate choice. Although we found strong evidence that both male and female behaviour contribute to mate choice in our trials, the facts remain that female genotypes were sorted linearly with respect to likelihood of mating, and that this linear sorting benefited males. The transitivity and repeatability of mate choice across different male genetic backgrounds and availability of signals show that the behaviour of both males and females is integral to mate choice, and male choice needs to be incorporated further into theoretical and empirical studies of sexual selection.

Our work is the first demonstration of transitive mate choice when individuals choose among repeatable genotypes. The vast majority of research in sexual selection relies on measuring the responses of individuals to one or few measured sexual traits, which we argue is not sufficient to evaluate the rationality of mate choice. Our data demonstrate that individuals make beneficial and hierarchical mate choices even when choosing among repeatable genotypes, rather than single sexual traits. Rational mate choice not only signifies mate choice's ability to

sort female genotypes in this species, but also demonstrates significant variation in female quality among potential mates' genotypes. Such genetic variation for female attractiveness and offspring production is a prerequisite for selection to act on female signals of quality and male preferences for them.

It is possible that the extent of the genetic variation in female quality was exaggerated by using inbred lines; however, these inbred lines were derived from a natural wild population, such that each allele present in our lines was present in the natural ancestral population, as was the genetic variation for female quality. It is also possible that the variation we observe in female attractiveness and productivity is the result of differing maturation rates among female genotypes, as we used relatively young (2–4 days old) female *D. melanogaster*. However, regardless of the underlying causes and the extent of variation in female quality, our data definitively demonstrate that males of many genotypes, including outbred populations, transitively sort female genotypes during mate choice, even among female genotypes of similar attractiveness rankings, to the benefit of these males. The fact that these inbred lines exhibit genetic variation for such evolutionarily important traits makes them a rich resource for future studies of the mechanisms underlying the maintenance of this variation. The fruit flies we tested can be used to assess potential costs of sexual attractiveness[23], trade-offs between sexually attractive traits and other life-history traits, and the condition dependence of attractive traits[30].

Lastly, we show that because males use multiple largely redundant sensory modalities to assess and interact with females, rational mate choices occur even with imperfect information. Males and females interact using several cues and these cues appear to communicate quality in a correlated manner. Such correlated sexual signals are consistent with a model in which a single developmental pathway (for example, insulin signalling) controls fecundity and multiple female traits relevant to mate choice[26,31], which would facilitate honest sexual signalling of female quality and promote rational advantageous mate choices. Phenotypes are complex and encompass several interacting individual traits; therefore, to understand mate choice, we need to understand how different traits interact to signal quality and how individuals process such cues. Our finding that fruit flies make rational mate choices when choosing among repeatable female genotypes based on redundant female cues greatly advances our understanding of how individuals assess potential mates and the variation in quality among individuals. Furthermore, rational mate choice has several far-reaching implications for the evolution of sexual signalling, the cognitive processes underlying mate choice, the maintenance of genetic variation, and the alignment of natural and sexual selection.

## Methods

**Fly stocks.** We obtained several isogenic lines of the DGRP[20], Canton-S and Oregon R laboratory stocks, and mutant $ninaB^{-/-}$ stocks from the Bloomington Stock Center. We used ten DGRP lines: 138, 313, 362, 399, 517, 555, 714, 738, 774 and 786. The $ppk23^{-/-}$ stock was generously provided by K. Scott. Canton-S and Oregon R stocks are outbred wild-type laboratory-adapted strains. The mutant $ninaB^{-/-}$ is functionally blind due to improper photoreceptor development[32]. The mutant $ppk23^{-/-}$ is deficient in the detection of female-specific contact pheromones[33].

For all experiments, we reared flies in standard cornmeal-sugar-yeast media. All flies were maintained at 24 °C and 50% relative humidity in a 12 h light:dark cycle. For all experiments, virgins were collected within 8 h of emergence under light $CO_2$ anaesthesia, with sexes held separately for 2–4 days at ten females or seven males per vial. For all mate choice assays, females were within 1 day in age of all other females.

**Mate choice assays.** To evaluate mate choice when males choose among DGRP females we carried out several separate mate choice assays. In all assays, we paired individual virgin females from 10 DGRP lines in all 45 possible pairwise line combinations. This design allowed us to evaluate the relative attractiveness of all female lines when in competition with any other line. Females were marked with yellow or red fluorescent powder (Brilliant Group, Inc.) 24–48 h before mating assays and introduced to experimental arenas 18–24 h before each mate choice assay. Females from each DGRP line were distributed randomly into all possible pairwise line combinations for that line. Trials were colour-balanced such that in each of the 45 pairwise combinations one line was yellow and its competitor red in half of the trials, whereas colours were switched in the other half. We initiated each mating trial by introducing a male into the experimental arena and observing it for up to 2 h. If mating occurred in this period, we noted the colour of the mated female. During mating assays, observers were blind to the line identity of females within each trial; thus, mated female line identity was recorded based on trial number and female colour after all observations had finished. Ten to 20 replicate trials were carried out for all 45 female combinations for each experimental male stock (see below for replicate numbers of each male tested and block). We measured the mate choice of each male only once, to control for mating experience, rather than measuring each individual's choices when exposed to a series of mating choices. We therefore measured mating preferences and transitivity via population or group-level 'voting'. Such measurement frameworks can mask the transitive choices of individuals if there is variation in the ordering of individual choices[26,27]. However, given the similarity in choices uncovered among several groups of unrelated males, it is unlikely that our data suffers from such distortions.

We measured wild-type male mate choice using the laboratory adapted Canton-S and Oregon R stocks. To measure mate choice in the absence of visual cues, we carried out mate choice trials with the blind mutant $ninaB^{-/-}$ (ref. 32) under normal light and wild-type Canton-S males under red light. *Drosophila* cannot see in the far red spectrum above 650 nm (ref. 34). To measure mate choice in the absence of specific female chemical cues, we used the mutant $ppk23^{-/-}$, which is deficient in the detection of female pheromones[33]. Lastly, to measure mate choice in the absence of both visual and chemical cues, we carried out mate choice assays with $ppk23^{-/-}$ males under red light. Although $ninaB^{-/-}$ and $ppk23^{-/-}$ mutants have different genetic backgrounds than either of the wild-type male populations, our previous assays demonstrated that unrelated males make very similar mating decisions, which allows us to compare the choices of mutant males with those of wild-type males. Furthermore, by carrying out mating trials in the dark, we are able to remove visual stimuli during mate choice, while controlling for genetic background for both Canton-S and $ppk23^{-/-}$ males.

Mate choice assays were carried out in several blocks. The following males were tested together: Canton-S assay 1 (14 replicates per pairwise female combination) and $ppk23^{-/-}$ assay 1 (20 replicates) were assayed over 2 separate 2-day blocks (Trial 1). Canton-S assay 2 (10 replicates), $ppk23^{-/-}$ assay 2 (12 replicates), $ninaB^{-/-}$ (14 replicates), Canton-S in the dark (14 replicates) and $ppk23^{-/-}$ in the dark (12 replicates) were carried out in a separate 2-day 2-block set (Trial 2). Oregon-R (ten replicates) was assayed separately in one single-day assay. In each assay, we maximized the number of replicates we could reliably observe.

**Mate choice data analysis.** Within a network, the relationship among three genotypes (A, B and C) is defined as a transitive triad if A > B, B > C and A > C, or an intransitive triad if A > B, B > C and C > A. Mate choice transitivity for an entire mating network, $t_{tri}$, was calculated for each male genetic background and treatment as the number of transitive triads divided by the total number of triads in the network (Fig. 1b), with greater scores signifying fewer intransitive inconsistencies[13]. We tested the significance of each network's transitivity by randomizing the direction of each relationship within the entire network 100 000 times and calculating the frequency with which randomized networks had transitivity scores equal to or greater than the observed $t_{tri}$.

Although assessing transitivity via triads is powerful and intuitive, the simplification of the network into a number of relationship subsets can miss patterns operating on the network as a whole[21,35,36]. We therefore validated our results via analysis of longer chains of relationships following Regenwetter et al.[21] We used the programme qtest[36] to test for violations of transitivity within the entire choice network. This programme assesses the likelihood that we observe the data given the assumption that the series of preferences among all alternatives match the narrow possible requirements for transitivity. This test uses transitivity of preference as the null hypothesis, with significant *P*-values signifying that choice is intransitive. We specified the series of preference values among the ten female DGRP lines based on the ranking of these lines, which in turn was based on the number of competitor lines each line 'beat' in direct competitions, evaluating these rankings separately for each male tested. We then tested the Linear Orders model in qtest, setting the majority value as 0.5. In agreement with our analyses via triads, none of the males tested significantly differed from a model of complete transitivity ($P > 0.45$ for all males tested).

We next assessed the strength of transitivity, which refers to the strength of the preferences among the relationships in the choice network, denoted as $P(a,b)$ for the preference for *a* over *b*. We have already established that our data meet the requirements of weak transitivity, where $a > b > c$ and all preferences are $\geq 0.5$. Moderate transitivity occurs if $P(a,c) \geq \min(P(a,b), P(b,c))$. Strong transitivity occurs if $P(a,c) \geq \max(P(a,b), P(b,c))$. We quantified the number of violations for moderate and strong transitivity for each male choice network. Overall, there

were few violations of moderate transitivity at 12–23% of triads violating this framework among the males tested. There were several violations of strong transitivity for most male choosers at 43–53% of triads violating this framework. Therefore, it appears that male fruit flies exhibit moderate transitivity strength when choosing mates.

We assessed similarity in mating biases between male choosers using Pearson's correlations between the proportion that a particular female DGRP line was mated for all 45 pairwise female combinations for one male chooser and the same proportion for a second male chooser. Results for each of the 45 pairwise DGRP line combinations were made comparable across male choosers by keeping the listed DGRP line order consistent; for example, results for competitions between lines 138 versus 313 for all males were always calculated as number of line 138 females mated divided by the total number of matings within this specific pairwise competition. We compared the mating biases of males with the wild-type (Canton-S) counterparts that were assayed concurrently with them (for example, $ppk23^{-/-}$ versus Canton-S assay 1, $ninaB^{-/-}$ versus Canton-S assay 2).

**Female mass.** During Trial 1, we froze virgin females from all ten DGRP lines at $-80\,^{\circ}\text{C}$ for 24 h in groups of five. After this, we measured the mass of 7–8 five-female groups for each DGRP line to the nearest 0.1 mg. All females measured for mass were reared and collected alongside the females used in the mate choice assays, such that any environmental factors were held constant across assays. Mass data were analysed via a one-way analysis of variance with female line identity as the dependent variable. Sample sizes were as follows: R138–7, R313–8, R362–7, R399–8, R517–7, R555–8, R714–8, R738–8, R774–8 and R786–8.

**Female CHCs.** During Trial 1 as well, we extracted female CHC samples from each of the ten DGRP lines concurrently with the mating trials, as CHCs can change throughout an individual's circadian rhythm. As with the mass data, females used for CHC collection were reared and collected alongside those used in the mating trials and mass collection. CHCs were extracted as previously described[37] for 14–20 females from each DGRP line. Individual females were placed in 100 µl of hexane for 3 min and vortexed for 1 min to remove CHCs from the cuticle. After this, females were removed from the hexane and discarded. Each sample was analysed via an Agilent Technologies 6890N gas chromatograph. Specific sample sizes were as follows: R138–18, R313–14, R362–19, R399–17, R517–19, R555–18, R714–20, R738–18, R774–18 and R786–16. We calculated the relative abundance of each of 29 analysed CHCs (Supplementary Fig. 1) by dividing the area under each peak by the area under all analysed peaks. CHCs data are often transformed (for example, logcontrasts) to reduce the dimensionality of such compositional data and these transformed axes are analysed to evaluate associations between particular chemical compositions and effects on fitness[38]. In such analyses, the line or family acts as the unit of replication[39]. With just ten female genotypes, we do not have the statistical power to perform such multidimensional analyses; thus, we examined correlations between each CHC and attractiveness ranking among the ten inbred lines based on the choice of Canton-S males. If any individual CHCs showed substantial correlations ($|r| > 0.5$), we explored the relationships between these individual CHCs and the mate choice of all males further.

**Male courtship and female receptivity.** We carried out detailed observations of mating interactions between females of 9 DGRP lines (line 399 did not produce enough females to be included) and either Canton-S or $ppk23^{-/-}$ males. For each observation, a single female was placed in a vial and allowed to acclimate for 18–24 h, after which a single male was placed in the vial. We recorded the timing of male courtship and mating in each vial, observing each vial for courtship for up to 2 h. If courtship was observed, we watched the vial for up to an additional 2 h for mating. We observed 20 male–female interactions for each female line by male combination over two blocks separated by 1 day (360 observations total). Female receptivity was measured via the time from male courtship to mating, with shorter times reflecting greater receptivity. This measure decouples male effort to mate from female receptivity to some extent by correcting for differences in time to courtship for all males. We did not measure female receptivity to ninaB males due to limited time and resources. Receptivity was analysed via a LRT of a fixed effects model with female courtship to mating time as the response variable, and female DGRP line and male line as fixed effects. We calculated each line's average receptivity as the maximum average courtship to mating time among the nine measured lines minus the average courtship to mating time for the line in question.

**Female offspring production.** Following the receptivity assay described above, we removed males from any vial in which mating occurred. We then allowed females to lay eggs for two consecutive 2-day periods, transferring individual females into new vials between these periods. Adult offspring were counted 18 days after transfer. Specific sample sizes are as follows: mated to Canton-S males R138–13, R313–9, R362–19, R517–8, R555–15, R714–15, R738–17, R774–19, R786–12; mated to ppk23 males R138–5, R313–6, R362–15, R517–4, R555–6, R714–6, R738–10, R774–13 and R786–4. Owing to an abundance of vials that produced no offspring (10% of females), the data were not normally distributed. We therefore ran generalized linear models with zero inflation and a negative binomial distribution, with offspring production as the response variable and female

DGRP line and male line as fixed effects, and performed an LRT to determine the effect of female line on offspring production.

**Female trait data analysis.** The relationship between male mating biases and female traits was assessed via correlations between the proportion that the first female DGRP line was mated and the difference between the female DGRP lines in the target trait (for example, line 1 mass–line 2 mass). This framework allows us to examine the mating decisions of males for each choice given, as mating biases might be more extreme when the trait differences between available females are more extreme. However, by calculating pairwise line differences in female traits, we generate 45 data points from ten measurements for each female trait. To avoid the resulting issues with pseudoreplication, we determined the significance of such correlations by randomizing our data 10,000 times and calculating the frequency with which randomized correlations had equal or greater (or lesser in the case of negative correlations) $r$-values. Relationships between average DGRP line female traits (offspring production, receptivity, mass and relative abundance of CHC 9 and 10) were assessed with Pearson's correlation. The presence of an outlier was tested using a Bonferroni outlier test on a linear model, using residuals for linear models with offspring production as the response variable and each other trait as a dependent variable, with all relationships being assessed separately.

To confirm our results from the analyses of female receptivity line means, we carried out additional analyses using randomizations of individual mating times. We did this because summary statistics could hide important differences in individual female receptivity when variance is high. For example, an individual female could have high receptivity, although she originates from a low average receptivity line. In addition, line average receptivity times ignore data from females that did not mate in the receptivity assays. Therefore, we randomly paired data from individuals of each line with each other line and calculated the difference in the time from courtship to mating to form predictions of which female should be chosen if the more receptive female tends to be chosen. If a female did not mate during observations, she was assigned the maximum time of 200 min. For each line comparison, we calculated the mean difference in receptivity time and calculated the correlation between these receptivity differences with mate choice, as we did for other female traits. These additional analyses using individual-level data confirmed our previous analysis of line means. For all males tested, the greater the difference in receptivity between female lines, the greater the preference that males showed towards the more receptive female (Pearson's correlation with significance tested via permutations, CS males: $r = -0.77$, $P < 0.00001$; CS males in the dark: $r = -0.68$, $P < 0.00001$; $ppk23^{-/-}$ males: $r = -0.56$, $P = 0.0002$).

To correct for the potential role of female receptivity in the mate choices of CS males, we calculated the residuals in the relationship between mating preferences for all pairwise comparisons between all lines and the difference in female receptivity between the same lines. These residuals indicate the extent to which preferences for a specific female line were greater or lesser than expected based on female receptivity alone for each female line combination. From these residuals, we reassigned the mate choice of males (line 1 was preferred if mating bias for this line was greater than expected based on receptivity alone, whereas line 2 was preferred if mating bias for this line was greater than expected). From this corrected preference matrix, we re-evaluated mate choice transitivity as described above.

**Male mate choice in the absence of female response.** Female attractiveness independent of female receptivity was tested using a two-choice behavioural assay, in which flies were video recorded and then videos were analysed using fly tracking software[27]. In this assay, two freshly decapitated 3-day-old virgin subject females were lightly embedded in solidifying agar 10–15 mm apart and 7–10 mm away from the side of the circular arena. After the agar solidified, a single 3-day-old virgin Canton-S male (previously isolated in individual microfuge tube with food for at least 2 h) was released in the arena and given 5–10 min to acclimate to the new environment. Video recording was then started and continued for 30 min. Videos were recorded at two frames per second and converted to AVI file format, which was analysed with VideoFly software that was developed in the Pletcher laboratory (freely available at http://sitemaker.umich.edu/pletcherlab/data). VideoFly calculates the amount of time spent by a focal fly inside a circle of 3 mm radius centred on each decapitated subject fly. Instances where the total time males spent in proximity to females was $< 50$ s were removed. Female attractiveness was calculated as the percentage of time males spent within a 3 mm radius of the more attractive female (as determined by live male choice assays previously) divided by the total time spent in proximity to both females. Pheromone transfer experiments have clearly demonstrated that male choice in these assays is largely based on female CHCs[26,27]. We carried out observations over two replicates for five female-line competitions each: R362 versus R517 (41 trials total), R362 versus R555 (27 trials), R 517 versus R138 (31 trials), R517 versus R313 (35 trials) and R774 versus R555 (24 trials). No difference was detected between two replicates for all five comparisons (by Wilcoxon test); therefore, the data from two replicates were combined for further analysis. For each of the five comparisons, we carried out Wilcoxin signed-rank tests to determine whether males showed significant preference for either female against the null hypothesis of no preference (50% time spent with the more attractive female). To test for the overall significance of the results for the five inactive female line

comparisons, we used a Fisher's combined probability test. For this test, we used $P$-values generated from one-tailed Wilcoxon signed-rank tests, with the null hypothesis being that males spend 50% or less of their time near the female previously determined as the less attractive line.

**Data availability.** All relevant data are available from D. Arbuthnott upon request.

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

## Acknowledgements

We thank Erin Tudor, Quynh Tran, Sharon Ornels, Alexandria McCarthy, Laurie Huang, William Gordon, Nick Force, Erika Gajda, Jake Mouser, Cindy Tseng and Eric Vanderbilt-Mathews for experimental help. Howard Rundle and Julie Colpitts performed gas chromatography for CHC extractions. Carly Ziegler provided valuable comments on an earlier draft. This work was funded by NIH grant GM102279 to S.D.P. and D.E.L.P and NIA Training Grant AG000114 to T.Y.F.

## Author contributions

All authors devised and planned experiments. D.A. and T.Y.F. carried out experiments. D.A. and D.E.L.P. analysed the data. All authors wrote the paper. S.D.P. and D.E.L.P. funded the work.

## Additional information

**Competing financial interests:** The authors declare no competing financial interests.

