## [Peer Review File · Nature Communications]

Reviewers' comments:

Reviewer #1 (Remarks to the Author):

This paper deals with a major issue in sexual selection. The question of the transitivity of the preference in non-human species has already been the subject of a quite large literature in a foraging context. But, surprisingly, the existence of transitive preferences in a sexual context has been very poorly assessed so far, despite their tremendous consequences in sexual selection. This is possibly due to the fact that the assessment of transitive preferences requires repeated sequential tests, which is a serious experimental limitation as most species are unable to mate repeatedly in a short period of time. The present MS is clearly a significant contribution to this debate. Yet, there are still major issues with the analysis reported. I recommend that the authors address these important concerns prior acceptance of the manuscript.

Major concerns.

1) My main concern is related to the fact that the transitivity of the sexual preference has not been assessed at the individual level but at the population level. The males were only tested once (in one single binary choice assay) and not several time in sequential assay. Thus, the transitive pattern reported herein is merely a pattern of group preference, something like the result of a vote at the level of the population (or lines) understudied. The introduction of the manuscript is written as a standard paper about transitivity and rational choice (whereas some caveats and uncited papers should be consider here, see below, point 3). Thus the authors knows full well that the transitive preference is fundamentally an individual property and should not be confused with majority preference, social choice and aggregation of vote at population level (see for instance Regenwetter et al 2009.

Behavioural social choice: a status report. Philosophical Transactions of the Royal Society B: Biological Sciences 364:833-843).

I guess that the experimental design used in this study is the consequence of strong biological constraints, the repeated copulations of the males introducing many bias (sperm depletion, chemical cues signalling non-virgin males and affecting the female choice, mate copying, etc.) in the analysis. Yet, I regret that the authors did not acknowledge this non-negligible limitation of their work. At the very least, I recommend that the authors discuss this interpretation issues in their manuscript. It is perfectly possible to observe social transitivity (at the line level) without assuming transitive preference at the individual level. Similarly, I recommend avoiding a constant confusion between social preference and individual preference throughout the MS. They should always prefer the word "the male line choice" instead of referring to "the male choice", which could be erroneously interpreted as the "individual male choice". For instance the sentence "male fruit flies exhibit rational choice" (lines 116) is ambiguous. At most, the authors observed "a male line fruit flies exhibit rational choice".

Having said this, I still consider that the observation of a transitive social preference is still an important result which deserve publication.

2) I regret that the transitivity was assessed at the level of the triads ($A > B > C$). The

protocol used by the authors allow the assessment of transitive preference on monger chain, which is a much powerful way to do it. Here, I believe that the paper by Regenwetter et al. (2010. Testing transitivity of preferences on two-alternative forced choice data. *Frontiers in Psychology* 1:148) would be of some help.

Similarly, I regret that the strength of the transitivity (weak VS. strong transitivity) is never discussed in the MS because this information is straightforward to compute from the data

3) In the introduction of the MS the authors referred to a quite old literature about rational choice. Some recent studies discussed the fact there is no contradiction between irrational (intransitive) choice and evolutionary optimal decision. For instance, I recommend the following papers:

- McNamara et al. 2014. Natural selection can favour 'irrational' behaviour. *Biology Letters* 10:20130935

- Hagen et al. 2012. Decision making: what can evolution do for us? pp. 97-126. In P. Hammerstein and J. R. Stevens (eds.), *Evolution and the mechanisms of decision making*, chapter 7. The MIT Press, Cambridge, Massachusetts.

- Fawcett et al. 2016. The evolution of decision rules in complex environments. *Trends in Cognitive Sciences* 18:153-161

4) I believe that one strength of this paper is related to the assessment of the fitness consequences of the transitive choice. Yet, I require that the authors properly test for this relationship. Instead of reporting the significant correlation between the fitness of a line and some cues possibly used by the males during mate choice, I recommend the authors to test directly for the relationship between the fitness of the female lines and the strength of the preference by the males.

Minor concerns.

Lines 115 and Fig.1. I am a little bit confused by what seems to be a contradictory result: $P_{tri} = 0.95$ (lines 115) or $P_{tri} = 1$ (Fig.1)?

The description of the protocol is very difficult to read. I recommend serious rewriting with a clearer and more logical presentation of the methods. For instance, the description of the control lines (*ninaB*^{-/-} and *ppk23*^{-/-}) is to be found in the "Fly stocks" paragraph. The paragraphs about "mate choice assays" and "mate choice analysis" are much easier to read when considered together.

Lines 273: "we combined DGRP virgin females". How much females in the same vial?

Lines 283: the report "10-20 replicate" deserves further explanations.

Lines 295: "our assays demonstrate that diverse genetic backgrounds interact with the female lines in a similar fashion". This sentence is highly unclear.

Lines 358-361: as the authors are studying count data with zero-inflation, I strongly recommend the use of generalized linear model with both negative binomials family link and

zero-inflation option (Zuur 2009. Mixed effects models and extensions in ecology with R. Springer, New-York). With R software, such analysis are conveniently performed using the 'glmmADMB' package.

Fig 1. I do not understand clearly how the permutations have been performed. Please explain.

Reviewer #2 (Remarks to the Author):

Review of: 'Male mate choice in fruit flies is rational and adaptive.'

The ms reports on testing the transitivity of mate choice by male flies, finding that they make transitive choices, and that these choices appear to be rational as the females they choose typically have more offspring. The results also suggest that there is some redundancy in the female cues (visual vs chemical).

In general, the ms is clear and well written. (I particularly liked the shrewd way, in the first para of the discussion, that the authors dealt with the possibility that results were generated by female choice rather than that of the males.)

My biggest doubt is whether there is sufficient novelty in the findings for it to be published by Nat Comms. The subject of transitivity has been looked at in numerous non-human species, from food choices in organisms as diverse as slime-moulds and birds, to nest-building decisions in ants. The authors also point out that transitivity in mate choice has also been looked at a couple of times in non-human species. Where then, is the real novelty? Given the number of species out there, the choice of flies seems of little import. So perhaps the main point of novelty that remains is the method of comparison. This is worth some thought on a couple of fronts. The authors suggest that the existing papers are attempting to discern a difference in mates that differ in just one trait (line 67), but this is not the case; Kirkpatrick et al. (2006, p.1220) point out that their frog calls differed in many parameters. Dechaume-Moncharmont et al. (2013) also look at the overall pattern, before concluding that there is a simple way of generating predictive power using a single-axis; I believe that this has confused the authors of this current ms. (The fundamental theory on this is that if the numerous traits cannot ultimately be mapped to a single preference function, then choices cannot always be transitive; to assume that they can - or that they cannot - would be to put the cart before the horse.) So that is not novel in this study. On a more positive front, this study makes use of inbreeding to compare the choices of individuals who are genetically identical - which does appear to be novel (line 215). Given the difficulty of controlling for changes in mate-preferences with experience, this seems very useful. Overall though, I have significant doubts about the worth of the ms for this journal.

General points for improvement:

Line 70-73: it's not the case that rational choice theory requires that information needs to

always be processed. (Imagine choosing between two houses; only if they are of very similar quality on the major points, like price, size and location, will the information on much smaller points need to be processed to reach a decision). Thus the point at line 143 is weakened so considerably that it becomes a non-sequitur -- as (line 141) although there's a strong correlation, it's not perfect.

In Figure 1A, there appears to be an arrow missing between R174 and R555; but perhaps this is a line of zero weight (no overall direction)? I found the width of the lines in my print-out largely indistinguishable, so would encourage the authors to magnify the effect if they are to use it, as mentioned in the caption.

I am confused by Fig 1C, which seems to indicate that the observations showed perfect transitivity; this seems to conflict with the text (e.g., line 115, 'significantly transitive'). Also, regarding Fig 1C, it is perhaps worth noting that we would expect 75% of results to appear transitive even if choices were random (as there are 8 ways of drawing preference arrows between three entities, only two of which are non-transitive), to help clue readers into the histogram. I'm not sure why 100,000 permutations were used (line 498) - or if this was 10,000 (line 395); I'd prefer a simple pdf if possible (presumably the frequency scale is meaningless as it's simply a function of the bin size?).

Small points:

In places, the authors talk about qualities varying 'continuously' (e.g., lines 38 and 222). Although, looking at the results, there are surely many levels, the language doesn't fit well with 'genotypes' (which although complex, are discrete), and the results certainly can't show that there is a true continuum.

Line 240: suggest changing 'be made' to 'often occur', due to the non-perfect correlation - and with it, perhaps on l.241 either cut 'redundant' or modify it to 'largely-redundant' or 'somesuch'.

In conjunction with the above, I suggest modifying the wording of the abstract slightly (e.g., 'act redundantly' to 'are significantly redundant'; 'persists' to 'largely persists').

I was slightly confused by lines 359-363. My reading of it is that it seems that 10% of females had no offspring (or was this a vial thing?), so the data were not normally distributed. You then removed those from the dataset, and found the other results were normally distributed. But you then say that the results were qualitatively the same whether you removed them or not. But the removal changed the qualitative result from not-normal to normal, so it seems to be a qualitative change.

Reviewer #3 (Remarks to the Author):

This paper reports an extensive and interesting set of behavioral experiments which the authors purport to show that the male mate preference for females is transitive and "rational". The authors use females from 10 defined inbred lines (from the PGRP collection) to stage binary choice trials, whereby males are confronted with two virgin females from different lines, and the female with which the male mates first is scored as being preferred. The main result of interest is that (1) this preference score turns out to be quite transitive (i.e., if females from line A are preferred to those from line B, and the latter are preferred to those from line C, in most cases A females will be preferred to C females; (2) the preference score is positively correlated with the difference in the lines in fecundity (i.e., rational). Additional assays show that choice scores in the absence of visual cues or by males with impaired olfaction are still quite well correlated to the choice scores by wildtype males able to use both types of cues. Additional assays explore correlations between the choice scores and other female phenotypes of the 10 lines (in addition to fecundity mentioned above, also the presence of specific cuticular hydrocarbons and female receptivity, quantified as latency to mating in 1 male - 1 female situation).

The question asked by this study is highly original and I am impressed by the amount of behavioral data the authors collected; it represents a massive amount of work. The results are interesting, but I am not convinced that the data fully supports the authors' interpretation.

The main - and a major - problem is that what the authors interpret as "male preference" - i.e., which of the two females mates first - is in fact ultimately the results of the female's decision to accept the male (females have the control over mating in *Drosophila*). So yes, it may indeed reflect the male's choice to focus his courtship effort on the preferred female, but it may just as well be the outcome of difference in between the females in their receptivity/eagerness to mate, with male being indiscriminate in his courtship. Arguably, the latter explanation is more parsimonious - as the authors emphasize, their putative finding of transitivity in male preference is novel certainly not a foregone conclusion; in contrast, if each line were characterized by a specific distribution of times to mating, the transitivity would be automatically achieved.

Unfortunately, the authors do not offer any data on male behavior that would allow one to assess male preference in a way that is less entangled with - even if not completely independent of - the female behavior. And, they do find that their measure of "male preference" is positively correlated across lines with female receptivity quantified in a separate experiment as latency to mate in one male - one female situation. In fact, under normal conditions (in light and with a wildtype male) this is the strongest of any correlation involving the "male preference" ($r = 0.81$, supplementary figure S2d). Yet, while the authors do acknowledge that alternative explanation, they dismiss it based on the argument that when the olfaction-impaired mutant *ppk23* males are used, the correlation between the "male preference" and female receptivity is "weak" (although still significant, $r = 0.3$), whereas the "male preference" of *ppk23* males is "highly" correlated with the "preferences" of wildtype CantonS males ($r = 0.68$). Thus, the authors argue that "variation in the importance of female receptivity in male mate choice indicates it is unlikely to be female choice alone that underlies our observed patterns" (I will return to the "alone qualifier

below"). By its very nature, this kind of argument is weak and too indirect to reject a simple explanation in favor of one that is more interesting but less parsimonious. Furthermore, I see a couple of problems with the way with the specific premises of this arguments in this case:

(i) To assess the association across the lines between "male choice" and female receptivity, the authors calculate the correlation between the male preference score between two female lines and the difference in mean latency to mate between these female lines. However, even if female receptivity completely determines which female mates first, one does not expect the mean latency to mate to be the best predictor of the outcome, especially if variance or distribution of individual latency to mate differ between the female lines. Consider e.g., two lines, where females of line A have mating latencies of 8,9,10,10,11,12 min (mean latency 10 min), whereas those of line B have latencies 5,6,6,7,20,28 min (mean latency 12 min). If these females were paired at random, female from line B would be expected to mate first in 2/3 of replicates, despite having a longer mean mating latency (i.e, lower mean receptivity). This kind of mismatch between the difference in the mean receptivity between lines and the sign of the corresponding difference between two individual females sampled from the two lines is likely greater when males come from the ppk23 strain because they generally mate more slowly. Therefore, rather than the current correlations in fig S2d, the authors should use the actual distributions of receptivity of individual females in each line to predict the male preference scores, under the assumption the latter are completely determined by the sign of the difference in individual receptivity values between the females. The correlation between these predicted and the actual preferences scores from the choice experiment could then be used to estimate the lower limit on the proportion of variation in choice scores explained by variation in female receptivity.

(ii) There were apparently many cases - more than half for the ppk23 males - where no mating occurred within the 2 hours of the receptivity assay. So there is a lot of censored data in that assay, but I found no information on how these were treated. Obviously, excluding them would result in a bias. Such censored observations would be straightforward to incorporate when predicting the outcome of "male choice" assays (i.e. It is not clear to me how to deal with them in the context of other types of correlations reported in the paper (e.g., between female receptivity and reproductive output).

These points of criticism concern the results interpreted by the authors as showing that the male choice is transitive, but similar issues can be raised about their claim that the choice is rational. In fact, the paper is somewhat confusing in this respect, because much of the text seems to imply the logic "rational choice is transitive, we see that male choice is transitive, therefore male choice must be rational", which is an obvious epistemological fallacy. Sometimes even "transitive" and "rational" seem to be used as synonyms (e.g., l. 127ff). "Rationality" is certainly something more difficult to define objectively than transitivity, but in the present context it would be most appropriate to define "rational" as something that maximized fitness. So it is only when the author show that the apparent male preference for one female genotype over another is positively correlated with the difference in their fecundity that I can see a reason to call this a rational choice. I think they should be clearer

about the distinction between the two concepts. But here, again, the problem is that this result may be entirely driven by variation among lines in female receptivity and the very strong correlation between receptivity and fecundity ($r = 0.91$). I note that the experiments used females that were 2-4 days old, i.e., not only very young, but also with large variation in the degree of maturation (*Drosophila* females take about 5 days to reach their full reproductive potential). It could be that the lines vary in the rate of maturation, females that have more eggs ready to fertilize are also more eager to mate, which could drive the observed pattern in the absence of any mate preference by the males.

I understand that addressing these issues is not going to be easy. However, I can see at least some possibilities:

(i) The authors should check statistically if the transitivity result and the relationship between "male choice" and female fecundity still holds if the difference in female receptiveness is statistically accounted for.

(ii) The authors actually do have data on at least one male behavior, namely the latency until the first courtship of a female by the male in one-to-one setting. If the male preference score is indeed driven by the female's attractiveness to the male, the time to first mating should be strongly negatively correlated with the preference scores (i.e., females from the preferred lines should be courted earlier).

(iii) The paper would gain in strength if the authors could obtain some data on the male behavior in the choice situation, such as which female is courted first, or what proportion of time the male spends courting one versus the other female. Of course, it would be unrealistic to expect the authors to do this for all pairs of lines. However, it should be possible for, let's say, three pairs of lines consisting of a highly preferred line and a line in the 3rd quartile of preference ranking (the very least preferred lines might have something wrong with them, given that these are inbred lines). If the authors' interpretation is right, the males should from the start bias their courtship to the female from a line with a higher preference score. In contrast, if the males court the females indiscriminately, the interpretation in terms of male choice would not be justified.

Finally, even if the authors do provide a more convincing evidence for male preference and its transitivity and rationality, the paper should acknowledge much more strongly the role of female receptivity. Currently there is a strange mismatch between the admission in the 1st para of the discussion that "it is unlikely that female choice alone" is responsible for the patterns observed, and the rest of the paper, which sounds like the authors have demonstrated unambiguously that which female mates first is essentially driven by male preference, with female behavior being irrelevant; this is particularly strong in the abstract. The findings should be presented in a more balanced way.

I understand that this is asking for a lot, but the paper is making quite an extraordinary claim which, as the authors well know, goes against what most people familiar with *Drosophila* would expect. E.g., according to the literature, naïve males (such as those used in this study) cannot even initially discriminate between mated (= unreceptive) and virgin

females, or between conspecific and heterospecific females - they court them indiscriminately and only acquire this ability to focus on virgin conspecifics through experience (there are several papers by Dukas showing this, see also Ejima et al 2007 Current Biol). So claiming that such naïve males can actually rank virgin females in a consistent way according to their fecundity requires very strong and unambiguous evidence.

Miscellaneous points:

- Were all females of a given line used in a given experiment all raised in one vial or in multiple vials, and if the latter, what is the effect of the vial on the variables reported? The lower-than-expected correlation in the outcome of the mating trials between the two experiments using CantonS males suggest that a large component which is due to a common environment rather than genotype.

- It is not clear what the trend lines in the correlation plots actually represent. If they are linear regression lines, this would be incorrect as regression assumes that the variable on the X-axis is measured without error (or at least with a much smaller error than those on the Y-axis). This is not the case here, and so the authors should plot the major axis (aka geometric mean) regression (e.g., Leng L, Zhang T, Kleinman L, Zhu W. 2007 Journal of Physics: Conference Series 78, 1-5.).

- Females aged 2-4 days were used. Did the age differ systematically between lines? Can the authors test for the effect of female age on the patterns observed? (See my comment on female state of maturation as a potential confounding factor above).

I. 145: from fig 2b, mating rate already dropped drastically when only the olfactory cues were removed.

Reviewer 1:

This paper deals with a major issue in sexual selection. The question of the transitivity of the preference in non-human species has already been the subject of a quite large literature in a foraging context. But, surprisingly, the existence of transitive preferences in a sexual context has been very poorly assessed so far, despite their tremendous consequences in sexual selection. This is possibly due to the fact that the assessment of transitive preferences requires repeated sequential tests, which is a serious experimental limitation as most species are unable to mate repeatedly in a short period of time. The present MS is clearly a significant contribution to this debate. Yet, there are still major issues with the analysis reported. I recommend that the authors address these important concerns prior acceptance of the manuscript.

Major concerns.

1) My main concern is related to the fact that the transitivity of the sexual preference has not been assessed at the individual level but at the population level. The males were only tested once (in one single binary choice assay) and not several time in sequential assay. Thus, the transitive pattern reported herein is merely a pattern of group preference, something like the result of a vote at the level of the population (or lines) understudied. The introduction of the manuscript is written as a standard paper about transitivity and rational choice (whereas some caveats and uncited papers should be consider here, see below, point 3). Thus the authors knows full well that the transitive preference is fundamentally an individual property and should not be confused with majority preference, social choice and aggregation of vote at population level (see for instance Regenwetter et al 2009. Behavioural social choice: a status report. Philosophical Transactions of the Royal Society B: Biological Sciences 364:833-843).

I guess that the experimental design used in this study is the consequence of strong biological constraints, the repeated copulations of the males introducing many bias (sperm depletion, chemical cues signalling non-virgin males and affecting the female choice, mate copying, etc.) in the analysis. Yet, I regret that the authors did not acknowledge this non-negligible limitation of their work. At the very least, I recommend that the authors discuss this interpretation issues in their manuscript. It is perfectly possible to observe social transitivity (at the line level) without assuming transitive preference at the individual level.

The reviewer is correct that we did not measure the transitivity of individual choice, and did not acknowledge this caveat. We now point out this issue (lines 136 - 142, 360 - 367).

“Though transitive choice is typically considered a property of individuals making a series of choices, we only tested each male once, in order to control for potential changes in mating behaviour through time and dependent on previous sexual interactions. We therefore were measuring population-level preferences and transitivity rather than individual-level transitivity. When discussing male mate choice in our results, we are referring to this group-level preference.”

However, while we note this caveat, we also point out that given the strong repeatability of mating preferences among unrelated groups, it is unlikely that our group-level measurements are masking different patterns operating at the individual level (lines 155 - 159).

“Though population-level, or “vote-based” measurements of choice can mask individual-level transitivity if different individuals have different transitive preferences^{26,27}, the striking similarity in the choices of these two unrelated populations of males suggests that our group-level assessment of choice reflects individual preferences and transitivity.”

Similarly, I recommend avoiding a constant confusion between social preference and individual preference throughout the MS. They should always prefer the word "the male line choice" instead of referring to "the male choice", which could be erroneously interpreted as the "individual male choice". For instance the sentence "male fruit flies exhibit rational choice" (lines 116) is ambiguous. At most, the authors observed "a male line fruit flies exhibit rational choice".

We have clarified the meaning of “male choice” within our results (lines 139 - 142).

“We therefore were measuring population-level preferences and transitivity rather than individual-level transitivity. When discussing male mate choice in our results, we are referring to this group-level preference.”

Having said this, I still consider that the observation of a transitive social preference is still an important result which deserve publication.

2) I regret that the transitivity was assessed at the level of the triads ($A > B > C$). The protocol used by the authors allow the assessment of transitive preference on monger chain, which is a much powerful way to do it. Here, I believe that the paper by Regenwetter et al. (2010. Testing transitivity of preferences on two-alternative forced choice data. *Frontiers in Psychology* 1:148) would be of some help.

We have performed additional analyses on our data to assess transitivity of longer chains alongside our analyses at the level of triads (described in Supplement lines 20 - 34). These additional analyses confirmed our previous analyses showing transitive mate choice in all males tested (lines 401 - 406).

“While assessing transitivity via triads is powerful and intuitive, the simplification of the network into a number of relationship subsets can miss patterns operating on the network as a whole^{26,37,38}. We therefore validated our results via analysis of longer chains of relationships following Regenwetter et al.²⁶. In all cases, these more detailed analyses yielded the same conclusions of transitivity for the mate choice of male fruit flies. See supplement for details.”

Similarly, I regret that the strength of the transitivity (weak VS. strong transitivity) is never discussed in the MS because this information is straightforward to compute from the data

We also performed additional analyses assessing the strength of transitivity in our data (Supplement lines 35 – 46).

“We next assessed the strength of transitivity, which refers to the strength of the preferences among the relationships in the choice network, denoted as $P(a,b)$ for the preference for a over b . We have already established that our data meet the requirements of weak transitivity, where $a > b > c$ and all preferences are 0.5 or greater. Moderate transitivity occurs if $P(a,c) \geq \min(P(a,b), P(b,c))$. Strong transitivity occurs if $P(a,c) \geq \max(P(a,b), P(b,c))$. We quantified the number of violations for moderate and strong transitivity for each male choice network. Overall, there were few violations of moderate transitivity at 12 – 23% of triads violating this framework among the males tested. There were several violations of strong transitivity for most male choosers at 43 – 53% of triads violating this framework. Therefore, it appears that male fruit flies exhibit moderate transitivity strength when choosing mates.”

3) In the introduction of the MS the authors referred to a quite old literature about rational choice. Some recent studies discussed the fact there is no contradiction between irrational (intransitive) choice and evolutionary optimal decision. For instance, I recommend the following papers:

- McNamara et al. 2014. Natural selection can favour 'irrational' behaviour. *Biology Letters* 10:20130935

- Hagen et al. 2012. Decision making: what can evolution do for us? pp. 97-126. In P. Hammerstein and J. R. Stevens (eds.), *Evolution and the mechanisms of decision making*, chapter 7. The MIT Press, Cambridge, Massachusetts.

- Fawcett et al. 2016. The evolution of decision rules in complex environments. *Trends in Cognitive Sciences* 18:153-161

We have expanded our introduction to include more recent work on evolutionary frameworks of decision making (lines 52 - 55).

“Recent theoretical work from an evolutionary perspective^{10,11} has also challenged the assumption that adaptive decision rules necessarily produce transitive choices, though these frameworks have yet to be thoroughly tested.”

(lines 81 – 87).

“In fact, recent theoretical work^{10,11} has suggested that preferences which violate transitivity may be adaptive in fluctuating environments. Given the complications discussed here, one cannot assume that observed linear preferences for single traits, which are typically the focus in sexual selection literature, will lead to transitive choices among potential mates when those mates express multiple sexual signals simultaneously.”

4) I believe that one strength of this paper is related to the assessment of the fitness consequences of the transitive choice. Yet, I require that the authors properly test for this relationship. Instead of reporting the significant correlation between the fitness of a line and some cues possibly used by the males during mate choice, I recommend the authors to

test directly for the relationship between the fitness of the female lines and the strength of the preference by the males.

Though we had directly assessed the correlations between male mating preference and the fitness consequences of such choices, this was not adequately emphasized in the previous version of this manuscript. We have therefore made the relationship between choice and female offspring production more apparent (Lines 210 – 215).

“More importantly, when this outlier line was removed, mate choice was significantly correlated with female productivity for all males tested (Fig. 3B). Therefore, male fruit flies’ mate choice matched our prediction that males make transitive mate choices by choosing the most productive female when exposed to any combination of females, qualifying males’ mate choices as rational.”

Minor concerns.

Lines 115 and Fig.1. I am a little bit confused by what seems to be a contradictory result: $P_{tri} = 0.95$ (lines 115) or $P_{tri} = 1$ (Fig.1)?

This confusion arose because the text reflected the results from individual assays, while the figure represents the amalgamated results from these two replicate assays. We have clarified this distinction (Lines 658 – 660).

“The data used to generate parts a) and c) are the combined results of the separate mate choice assays using Canton-S males, while the statistics reported in the text were calculated for each assay separately.”

The description of the protocol is very difficult to read. I recommend serious rewriting with a clearer and more logical presentation of the methods. For instance, the description of the control lines (*ninaB*^{-/-} and *ppk23*^{-/-}) is to be found in the "Fly stocks" paragraph. The paragraphs about "mate choice assays" and "mate choice analysis" are much easier to read when considered together.

We have followed the reviewer’s recommendation and now describe all *Drosophila* lines in the “fly stocks” section of the methods, and the description of the mate choice data analysis now immediately follows the description of the mate choice assay methods.

Lines 273: "we combined DGRP virgin females". How much females in the same vial?

We have clarified that only two females were placed in each vial (lines 344 – 345).

“In all assays, we paired individual virgin females from 10 DGRP lines in all 45 possible pairwise line combinations.”

Lines 283: the report "10-20 replicate" deserves further explanations.

We now give full sample size details (lines 382 – 389).

Lines 295: "our assays demonstrate that diverse genetic backgrounds interact with the female lines in a similar fashion". This sentence is highly unclear.

We have clarified this sentence (lines 376 – 379).

"While $ninaB^{-/-}$ and $ppk23^{-/-}$ mutants have different genetic backgrounds than either of the wild-type male populations, our previous assays demonstrated that unrelated males make very similar mating decisions, which allows us to compare the choices of mutant males with those of wild-type males."

Lines 358-361: as the authors are studying count data with zero-inflation, I strongly recommend the use of generalized linear model with both negative binomials family link and zero-inflation option (Zuur 2009. Mixed effects models and extensions in ecology with R. Springer, New-York). With R software, such analysis are conveniently performed using the 'glmmADMB' package.

We re-analyzed our female productivity data as the reviewer recommended, which did not alter our conclusion of significant differences among inbred lines in offspring production (lines 478 – 482).

"Due to an abundance of vials that produced no offspring (10% of females), the data were not normally distributed. We therefore ran generalized linear models with zero inflation and a negative binomial distribution, with offspring production as the response variable and female DGRP line and male line as fixed effects, and performed an LRT to determine the effect of female line on offspring production."

Fig 1. I do not understand clearly how the permutations have been performed. Please explain.

We have clarified the permutation procedures in the methods section (lines 397 – 400).

"We tested the significance of each network's transitivity by randomizing the direction of each relationship within the entire network 100,000 times and calculating the frequency with which randomized networks had transitivity scores equal to or greater than the observed t_{tri} ."

Reviewer 2:

The ms reports on testing the transitivity of mate choice by male flies, finding that they make transitive choices, and that these choices appear to be rational as the females they choose typically have more offspring. The results also suggest that there is some redundancy in the female cues (visual vs chemical).

In general, the ms is clear and well written. (I particularly liked the shrewd way, in the first para of the discussion, that the authors dealt with the possibility that results were generated by female choice rather than that of the males.)

My biggest doubt is whether there is sufficient novelty in the findings for it to be published by Nat Comms. The subject of transitivity has been looked at in numerous non-human species, from food choices in organisms as diverse as slime-moulds and birds, to nest-building decisions in ants. The authors also point out that transitivity in mate choice has also been looked at a couple of times in non-human species. Where then, is the real novelty? Given the number of species out there, the choice of flies seems of little import.

Though transitivity of choice has been looked at in terms of food or habitat choice, it has been almost completely ignored in terms of mate choice. The novelty of our set of experiments is that by using individuals with known genotypes, we can assess mate choice at the whole-individual level, rather than making conclusions based on a subset of traits. Therefore, our findings of mate choice transitivity, as well as redundancy in female traits targeting different sensory modalities is novel, and has important implications for the field of sexual selection and animal behaviour.

So perhaps the main point of novelty that remains is the method of comparison. This is worth some thought on a couple of fronts. The authors suggest that the existing papers are attempting to discern a difference in mates that differ in just one trait (line 67), but this is not the case; Kirkpatrick et al. (2006, p.1220) point out that their frog calls differed in many parameters.

We clarify that, while the song traits of Kirkpatrick et al. encompass several song parameters, the study still relies only on mate choice assessment using one sensory modality (lines 70 - 75).

"Of the few tests that have been done on mate choice rationality¹⁴⁻¹⁶, each presented females with potential mates that differed in traits affecting just one sensory modality. Yet sexual displays are often complex, signaling to several different sensory modalities and differing in many ways within modalities¹⁷, and individuals of both sexes are faced with making mating decisions based on all of the information available, rather than on single traits in isolation."

Dechaume-Moncharmont et al. (2013) also look at the overall pattern, before concluding that there is a simple way of generating predictive power using a single-axis; I believe that this has confused the authors of this current ms. (The fundamental theory on this is that if the numerous traits cannot ultimately be mapped to a single preference function, then choices cannot always be transitive; to assume that they can - or that they cannot - would be to put the cart before the horse.) So that is not novel in this study.

Dechaume-Moncharmont et al. assessed mate choice in cichlids by giving females binary choices, as we did in our study. However, the authors of this study separated

males into triplets (from which two males would be drawn for binary choices) based on size alone (small, medium, and large). Therefore, the authors of this study assessed transitivity by presenting females with males that were chosen based on their difference in only one trait: size.

On a more positive front, this study makes use of inbreeding to compare the choices of individuals who are genetically identical - which does appear to be novel (line 215). Given the difficulty of controlling for changes in mate-preferences with experience, this seems very useful. Overall though, I have significant doubts about the worth of the ms for this journal.

As the reviewer notes and we outline above, our methods are novel, and the main novelty of our study lies in the power of our conclusions based on these methods.

Line 70-73: it's not the case that rational choice theory requires that information needs to always be processed. (Imagine choosing between two houses; only if they are of very similar quality on the major points, like price, size and location, will the information on much smaller points need to be processed to reach a decision). Thus the point at line 143 is weakened so considerably that it becomes a non-sequitur -- as (line 141) although there's a strong correlation, it's not perfect.

We have edited these sections to more accurately reflect the relevant theory and aims of our study, as the reviewer notes (lines 75 - 79).

"In fact, rational choice theory assumes that individuals have complete information about their choices¹ and that they may need to process all information available to choose the option with the greatest cumulative benefits if two options are very similar along several axes of assessment³."

(lines 160 - 165).

"Having demonstrated repeatable transitivity in male mate choice, we next characterized the nature of the information used to make such rational choices. To determine whether males exhibit transitive preference with limited information and whether female traits signal redundant or orthogonal components of sexual fitness, we measured male mate choice when sensory perception was impaired, manipulating sight, smell/taste, or both."

In Figure 1A, there appears to be an arrow missing between R174 and R555; but perhaps this is a line of zero weight (no overall direction)? I found the width of the lines in my print-out largely indistinguishable, so would encourage the authors to magnify the effect if they are to use it, as mentioned in the caption.

We have clarified the figure caption to explain missing arrows (lines 649 - 650).

"Absent arrows (e.g. R517 vs. R313) denote ties between two lines."

It appears that in converting our manuscript to pdf, the visual differences in the weight of the arrows in Fig. 1a was decreased. If our manuscript is accepted for publication, we will provide a high-resolution version of this figure separately, rather than embedding it in the manuscript file.

I am confused by Fig 1C, which seems to indicate that the observations showed perfect transitivity; this seems to conflict with the text (e.g., line 115, 'significantly transitive').

As we outlined above for Reviewer 1's comment, we have clarified this point of confusion in the caption of Fig. 1.

Also, regarding Fig 1C, it is perhaps worth noting that we would expect 75% of results to appear transitive even if choices were random (as there are 8 ways of drawing preference arrows between three entities, only two of which are non-transitive), to help clue readers into the histogram.

We have added this explanation to the figure (lines 654 – 656).

"On average, we expect that 75% of relationships would be transitive at random, as 6/8 possible relationships within a triad are transitive."

I'm not sure why 100,000 permutations were used (line 498) - or if this was 10,000 (line 395); I'd prefer a simple pdf if possible (presumably the frequency scale is meaningless as it's simply a function of the bin size?).

100,000 permutations were carried out (10,000 were used for permutations analysing the relationships between female traits and male preference). The standard error for p-values estimated from permutation studies is $\sqrt{(p(1-p)/N)}$ (Roff 2006) where p is the nominal p-value and N is the number of permutations. We choose 100,000 permutations to enable us to accurately assess the significance of our observed transitivity at a P-value ≤ 0.0001 , thus giving us greater confidence in our conclusions regarding the striking level of transitivity in male mate choice. We have replotted the figure using frequencies rather than counts, as suggested by the reviewer (Fig. 1c).

In places, the authors talk about qualities varying 'continuously' (e.g., lines 38 and 222). Although, looking at the results, there are surely many levels, the language doesn't fit well with 'genotypes' (which although complex, are discrete), and the results certainly can't show that there is a true continuum.

We have changed our language to more accurately reflect the data (e.g. lines 285 – 287).

"Rational mate choice not only signifies males' ability to evaluate and sort female genotypes, but also demonstrates significant variation in female quality among potential mates' genotypes."

Line 240: suggest changing 'be made' to 'often occur', due to the non-perfect correlation - and with it, perhaps on l.241 either cut 'redundant' or modify it to 'largely-redundant' or somesuch.

In conjunction with the above, I suggest modifying the wording of the abstract slightly (e.g., 'act redundantly' to 'are significantly redundant'; 'persists' to 'largely persists').

We have made all of these suggested changes (lines 34 – 37, 307 – 309).

I was slightly confused by lines 359-363. My reading of it is that it seems that 10% of females had no offspring (or was this a vial thing?), so the data were not normally distributed. You then removed those from the dataset, and found the other results were normally distributed. But you then say that the results were qualitatively the same whether you removed them or not. But the removal changed the qualitative result from not-normal to normal, so it seems to be a qualitative change.

Because we now analyse all of the data, including the overabundance of zeroes, following Reviewer 1's suggestion, the statement causing confusion has been removed.

Reviewer 3

This paper reports an extensive and interesting set of behavioral experiments which the authors purport to show that the male mate preference for females is transitive and "rational". The authors use females from 10 defined inbred lines (from the PGRP collection) to stage binary choice trials, whereby males are confronted with two virgin females from different lines, and the female with which the male mates first is scored as being preferred. The main result of interest is that (1) this preference score turns out to be quite transitive (i.e., if females from line A are preferred to those from line B, and the latter are preferred to those from line C, in most cases A females will be preferred to C females; (2) the preference score is positively correlated with the difference in the lines in fecundity (i.e., rational). Additional assays show that choice scores in the absence of visual cues or by males with impaired olfaction are still quite well correlated to the choice scores by wildtype males able to use both types of cues. Additional assays explore correlations between the choice scores and other female phenotypes of the 10 lines (in addition to fecundity mentioned above, also the presence of specific cuticular hydrocarbons and female receptivity, quantified as latency to mating in 1 male - 1 female situation).

The question asked by this study is highly original and I am impressed by the amount of behavioral data the authors collected; it represents a massive amount of work. The results are interesting, but I am not convinced that the data fully supports the authors' interpretation.

The main - and a major - problem is that what the authors interpret as "male preference" - i.e., which of the two females mates first - is in fact ultimately the results of the female's

decision to accept the male (females have the control over mating in *Drosophila*). So yes, it may indeed reflect the male's choice to focus his courtship effort on the preferred female, but it may just as well be the outcome of difference in between the females in their receptivity/eagerness to mate, with male being indiscriminate in his courtship. Arguably, the latter explanation is more parsimonious - as the authors emphasize, their putative finding of transitivity in male preference is novel certainly not a foregone conclusion; in contrast, if each line were characterized by a specific distribution of times to mating, the transitivity would be automatically achieved. Unfortunately, the authors do not offer any data on male behavior that would allow one to assess male preference in a way that is less entangled with - even if not completely independent of - the female behavior.

We have carried out additional analyses and behavioural assays which remove female behaviour from male choice entirely. First, at the reviewer's suggestion, we analysed the time to courtship in single male-female interactions, and found that males do indeed tend to show a weak but significant trend to court females faster if the female line was previously shown to be more attractive in two-female assays (Fig. 4a, lines 225 - 234).

*"To clarify the relative roles of male and female behaviour during our male choice trials, we carried out additional analyses and experiments. First, we exposed individual males (Canton-S and *ppk23^{-/-}*) to single females of each genotype and measured the lag time between male introduction and courtship, which is a proxy for male motivation to mate these females. We found a weak but significant negative correlation between time to courtship and male mate choice in two-female choice experiments for each male tested (Fig. 4a). This result suggests that, on average, the faster that males court females in a no-choice scenario, the more likely they are to choose that female's genotype in a choice experiment."*

More importantly, we carried out additional assays to assess male mate choice in the absence of female behaviour. To do this, we gave males the choice between two decapitated females from a subset of the female line combinations. We found that, for the most part, males showed similar preferences toward female lines whether females are active or not, suggesting that male choice is important in our live-female assays, though we did find evidence that female receptivity plays some role in the outcome of our assays for a small subset of female mate competitions (Fig. 4b, lines 235 - 247).

"Though latency to courtship is thought to be indicative of male choice, it can also be influenced by female receptivity and motivation to mate. We therefore carried out additional assays to observe male preferences in the absence of female behaviour. To do this, we measured the time males spent in proximity to two decapitated (and therefore non-responding) females from a subset of the previously used female line combinations. We found significant or marginally non-significant male preference for the more attractive (as determined in live assays) female genotype in four out of five tested female comparisons (Fig. 4b), and one comparison yielded no significant preference for either female. This demonstrates that male choice plays a significant role in our live mate choice assays. The fact that in one out of five tested female combinations male choice between decapitated females

did not repeat live choice suggests that in some instances female behaviour (receptivity) could strongly influence the outcome of mate choice."

And, they do find that their measure of "male preference" is positively correlated across lines with female receptivity quantified in a separate experiment as latency to mate in one male - one female situation. In fact, under normal conditions (in light and with a wildtype male) this is the strongest of any correlation involving the "male preference" ($r = 0.81$, supplementary figure S2d). Yet, while the authors do acknowledge that alternative explanation, they dismiss it based on the argument that when the olfaction-impaired mutant *ppk23* males are used, the correlation between the "male preference" and female receptivity is "weak" (although still significant, $r = 0.3$), whereas the "male preference" of *ppk23* males is "highly" correlated with the "preferences" of wildtype CantonS males ($r = 0.68$). Thus, the authors argue that "variation in the importance of female receptivity in male mate choice indicates it is unlikely to be female choice alone that underlies our observed patterns" (I will return to the "alone qualifier below"). By its very nature, this kind of argument is weak and too indirect to reject a simple explanation in favor of one that is more interesting but less parsimonious.

The reviewer is correct that our previous argument was indirect. Given the additional data we collected, we have removed this argument, and replaced it with data from our male choice experiments with inactive females (Lines 256 - 265).

"One could argue that the association we found between female receptivity and male choice ranking among the female genotypes indicates that it is female behaviour and choice rather than male choice that underlies mate choice transitivity in our data. However, when we carried out assays that removed female response behaviours from sexual interactions, in most instances, we found agreement in male choice between two female lines whether females were responding (unmanipulated) or non-responding (decapitated). These results suggest that female behaviour does not generally determine male mating decisions, and final choice is driven largely by male choice, though female receptivity likely plays an important role for a small subset of mating decisions."

Furthermore, I see a couple of problems with the way with the specific premises of this arguments in this case:

(i) To assess the association across the lines between "male choice" and female receptivity, the authors calculate the correlation between the male preference score between two female lines and the difference in mean latency to mate between these female lines. However, even if female receptivity completely determines which female mates first, one does not expect the `_mean_` latency to mate to be the best predictor of the outcome, especially if variance or distribution of individual latency to mate differ between the female lines. Consider e.g., two lines, where females of line A have mating latencies of 8,9,10,10,11,12 min (mean latency 10 min), whereas those of line B have latencies 5,6,6,7,20,28 min (mean latency 12 min). If these females were paired at random, female from line B would be expected to mate first in 2/3 of replicates, despite having a longer mean mating latency (i.e, lower mean receptivity). This kind of mismatch between the

difference in the mean receptivity between lines and the sign of the corresponding difference between two individual females sampled from the two lines is likely greater when males come from the ppk23 strain because they generally mate more slowly. Therefore, rather than the current correlations in fig S2d, the authors should use the actual distributions of receptivity of individual females in each line to predict the male preference scores, under the assumption the latter are completely determined by the sign of the difference in individual receptivity values between the females. The correlation between these predicted and the actual preferences scores from the choice experiment could then be used to estimate the lower limit on the proportion of variation in choice scores explained by variation in female receptivity.

We re-analysed our female receptivity as the reviewer suggested. These reanalyses provided the same conclusions as those resulting from analyses of female line means (Supplement lines 85 - 102).

(ii) There were apparently many cases - more than half for the ppk23 males - where no mating occurred within the 2 hours of the receptivity assay. So there is a lot of censored data in that assay, but I found no information on how these were treated. Obviously, excluding them would result in a bias. Such censored observations would be straightforward to incorporate when predicting the outcome of "male choice" assays (i.e. It is not clear to me how to deal with them in the context of other types of correlations reported in the paper (e.g., between female receptivity and reproductive output).

The reanalysis of the receptivity following the reviewer's suggestion has taken these previously censored data into account and includes them to generate predictions regarding male mate choice among female genotypes. Specifically, we gave a high time value to those females that did not mate, allowing for comparisons between females that mated and those that did not, as well as between females that did not mate.

These points of criticism concern the results interpreted by the authors as showing that the male choice is transitive, but similar issues can be raised about their claim that the choice is rational. In fact, the paper is somewhat confusing in this respect, because much of the text seems to imply the logic "rational choice is transitive, we see that male choice is transitive, therefore male choice must be rational", which is an obvious epistemological fallacy. Sometimes even "transitive" and "rational" seem to be used as synonyms (e.g., l. 127ff). "Rationality" is certainly something more difficult to define objectively than transitivity, but in the present context it would be most appropriate to define "rational" as something that maximized fitness. So it is only when the author show that the apparent male preference for one female genotype over another is positively correlated with the difference in their fecundity that I can see a reason to call this a rational choice. I think they should be clearer about the distinction between the two concepts.

We have edited several sections to ensure that we only use the word rational to describe our data when it relates to the fitness advantages of transitive choices.

But here, again, the problem is that this result may be entirely driven by variation among lines in female receptivity and the very strong correlation between receptivity and fecundity ($r = 0.91$). I note that the experiments used females that were 2-4 days old, i.e., not only very young, but also with large variation in the degree of maturation (*Drosophila* females take about 5 days to reach their full reproductive potential). It could be that the lines vary in the rate of maturation, females that have more eggs ready to fertilize are also more eager to mate, which could drive the observed pattern in the absence of any mate preference by the males.

We have included this as a possible mechanism underlying our results (lines 293 – 300).

*“It is also possible that the variation we observe in female attractiveness and productivity is the result of differing maturation rates among female genotypes, as we used very young (two to four days old) female *D. melanogaster*. However, regardless of the underlying causes and the extent of variation in female quality, our data definitively demonstrate that males of many genotypes, including outbred populations, transitively sort female genotypes during mate choice, even among female genotypes of similar attractiveness rankings, to the benefit of these males.”*

I understand that addressing these issues is not going to be easy. However, I can see at least some possibilities:

(i) The authors should check statistically if the transitivity result and the relationship between "male choice" and female fecundity still holds if the difference in female receptiveness is statistically accounted for.

(ii) The authors actually do have data on at least one male behavior, namely the latency until the first courtship of a female by the male in one-to-one setting. If the male preference score is indeed driven by the female's attractiveness to the male, the time to first mating should be strongly negatively correlated with the preference scores (i.e., females from the preferred lines should be courted earlier).

(iii) The paper would gain in strength if the authors could obtain some data on the male behavior in the choice situation, such as which female is courted first, or what proportion of time the male spends courting one versus the other female. Of course, it would be unrealistic to expect the authors to do this for all pairs of lines. However, it should be possible for, let's say, three pairs of lines consisting of a highly preferred line and a line in the 3rd quartile of preference ranking (the very least preferred lines might have something wrong with them, given that these are inbred lines). If the authors' interpretation is right, the males should from the start bias their courtship to the female from a line with a higher preference score. In contrast, if the males court the females indiscriminately, the interpretation in terms of male choice would not be justified.

To address this reviewer's concern, we chose to analyse male time to courtship, and to carry out experiments directly testing for male choice in the absence of female

behaviour, as we outlined above. Both of these additional components provide strong evidence for male choice playing a large role in our previous experiments. We have also added a paragraph to the introduction (lines 102 - 115) highlighting previous research on male mate choice, and its prevalence and importance in a number of species' mating systems, including *Drosophila melanogaster*.

Finally, even if the authors do provide a more convincing evidence for male preference and its transitivity and rationality, the paper should acknowledge much more strongly the role of female receptivity. Currently there is a strange mismatch between the admission in the 1st para of the discussion that "it is unlikely that female choice alone" is responsible for the patterns observed, and the rest of the paper, which sounds like the authors have demonstrated unambiguously that which female mates first is essentially driven by male preference, with female behavior being irrelevant; this is particularly strong in the abstract. The findings should be presented in a more balanced way.

In association with discussion of the new data demonstrating the males' role in mate choice, we have included further discussion of the role female behaviour plays (Lines 243 - 247).

"This demonstrates that male choice plays a significant role in our live mate choice assays. The fact that in one out of five tested female combinations male choice between decapitated females did not repeat live choice suggests that in some instances female behaviour (receptivity) could strongly influence the outcome of mate choice."

Lines 274 - 278

"The transitivity and repeatability of mate choice across different male genetic backgrounds and availability of signals, shows that the behaviour of both males and females is integral to mate choice, and male choice needs to be incorporated further into theoretical and empirical studies of sexual selection."

We have also changed our language throughout the paper to ensure that we do not overstate males' role in mate choice (e.g. Lines 37 - 40, 244 - 247, 262 - 265, 274 - 278).

I understand that this is asking for a lot, but the paper is making quite an extraordinary claim which, as the authors well know, goes against what most people familiar with *Drosophila* would expect. E.g., according to the literature, naïve males (such as those used in this study) cannot even initially discriminate between mated (= unreceptive) and virgin females, or between conspecific and heterospecific females - they court them indiscriminately and only acquire this ability to focus on virgin conspecifics through experience (there are several papers by Dukas showing this, see also Ejima et al 2007 Current Biol). So claiming that such naïve males can actually rank virgin females in a consistent way according to their fecundity requires very strong and unambiguous evidence.

Miscellaneous points:

- Were all females of a given line used in a given experiment all raised in one vial or in multiple vials, and if the latter, what is the effect of the vial on the variables reported? The lower-than-expected correlation in the outcome of the mating trials between the two experiments using CantonS males suggest that a large component which is due to a common environment rather than genotype.

Females were collected from a large number of vials, though some females were collected from the same vial. Given the large number of females collected and handled, we did not record each female's vial of origin, and therefore cannot assess any vial effect. However, given that there were strong line effects for each female trait measured, and because females were collected from a large number of vials, such vial effects are unlikely to have significantly affected our results.

It is not clear what the trend lines in the correlation plots actually represent. If they are linear regression lines, this would be incorrect as regression assumes that the variable on the X-axis is measured without error (or at least with a much smaller error than those on the Y-axis). This is not the case here, and so the authors should plot the major axis (aka geometric mean) regression (e.g., Leng L, Zhang T, Kleinman L, Zhu W. 2007 Journal of Physics: Conference Series 78, 1-5.).

We now specify that the major axis regression is represented in each of the figure legends.

Females aged 2-4 days were used. Did the age differ systematically between lines? Can the authors test for the effect of female age on the patterns observed? (See my comment on female state of maturation as a potential confounding factor above).

Females were all within one day in age of each other in each assay. We now make this clear in our manuscript (lines 339 – 340).

"For all mate choice assays, females were within one day in age of all other females."

As with the vial of origin, we were not able to keep track of each female's age in each trial, and instead sought to control age by ensuring all females were of similar age on each day of each assay.

l. 145: from fig 2b, mating rate already dropped drastically when only the olfactory cues were removed.

The drop in mating rate is likely largely influenced by our use of a mutant line. For example, both Canton-S males tested in the dark and ninaB mutants lacked the same visual cues, but the mutant ninaB line showed a much reduced mating rate. The key comparisons in mating rate can be seen when one male line is compared with and without a specific sensory modality (Canton-S vs. Canton-S in the dark, ppk23 vs. ppk23 in the dark).

Reviewers' comments:

Reviewer #1 (Remarks to the Author):

I have carefully read this new version of the MS, and I have to confess that, as a referee, I am impressed by the efforts made by the authors to address the reviewers's concerns. They not only modified the main text, discussed additional references, performed new (and much stronger) statistical analysis, but they also did complementary experiments. For instance, the new experiments (no choice experiment and decapitated females experiment) convincingly address the confounding effect of the female receptivity, and mutual mate choice. I consider that this manuscript is an important contribution (and a huge piece of work) which deserves publication in a high-impact factor journal.

My only minor recommendation would be to explain more carefully the non-trivial link between fitness and rationality. I believe that the excellent book chapter by Alex Kacelnick (2006. Meanings of rationality, pp. 87-105. Oxford: Oxford University Press) or the review by the MAD group (Fawcett, T. W., B. Fallenstein, A. D. Higginson, A. I. Houston, D. E. W. Mallpress, P. C. Trimmer, and J. M. McNamara. 2016. The evolution of decision rules in complex environments. Trends in Cognitive Sciences 18, 153-161) would be very helpful for the reader. I ask the authors to spend few lines to explain the concept of "ecological rationality" which is truly very different from the common sense of "rationality" in psychology for instance.

Reviewer #2 (Remarks to the Author):

I am happy with the changes that the authors have made, and look forward to seeing it published.

Reviewer #3 (Remarks to the Author):

I have mixed feelings about this revised version of the ms.

On the one hand, I can see that the authors made a serious effort to address my comments, as well as those by the other reviewers. In particular, as I suggested, they changed the way they analyze transitivity and the correlation between female receptivity and the outcome of the mate choice assays. They added a correlation between time to first courtship in a one-to-one situation and the outcome of mate choice assays. And they performed, for a few pairs of female lines, a new experiment with male choice between headless females, thus eliminating female response. These last two results now show that male preference plays at least some role in the observed phenomenon. This is good, as no such evidence was present in the previous version.

On the other hand, I do not find that the authors have sufficiently addressed my criticism of

their biased interpretation of the results as being entirely (or almost entirely) driven by male preference for females. The title, abstract, introduction and most of the result section are still written in terms of the results being unambiguously attributable to male mate choice, whereas in fact most of the assays reported (e.g. those described in l. 143 – 195) offer no possibility to distinguish the effect of male choice from those of female receptivity. Essentially, someone with as much bias for looking at female agency as these authors have for male agency could use these data to write a paper about the transitivity of female competitive prowess when competing for males. Although, as I said in the preceding paragraph, a couple of new results do support a role of male behavior, other results (notably the strong correlations between female receptivity in one-to-one trials and the outcome of "mate choice" trials) point to a (possibly greater) role of female behavior. This is especially the case with the new analysis, which shows such a correlation even for pkk23 males. I cannot avoid the impression that the authors are trying to sweep the inconvenient results under the carpet. E.g., despite their key importance for the interpretation of the mating trials, the correlations between female receptivity and the putative "male choice" is only described in supplementary material; in the Results section it is only alluded to in four (!) words (l. 192), and the potential role played by female behavior is not mentioned at all in the abstract.

The authors do express a more balanced view in the second paragraph of the discussion (l. 266ff), where they do admit that actually admit that what they found is that "female genotypes are sorted linearly with respect to the likelihood of mating", and while influence of both male and female behavior can be inferred, their relative importance remains unknown. This is good, but it means that there is currently a disconnect between this paragraph and the rest of the paper. The balanced attitude of this one paragraph should be extended to the entire paper. I believe an unbiased way of presenting and interpreting the results requires that the results of the 1 male/2 females mating trials be described in terms that are neutral with respect to male versus female agency. Finding the transitivity in these outcomes should then lead to the question of the relative contribution of male versus female behavior and the experiments that throw light on one versus another.

Furthermore, I reiterate my request for an analysis that reveals the degree to which the transitivity of the outcome of the putative "male choice" remains detectable after female receptivity is statistically controlled for. The authors argue in their response letter that they did the trials with headless females instead. However, this new experiment, while highly useful, only involves a subset of lines (understandably, as this is a large effort), and so it does not directly address the transitivity. I do not see why this new experiment should be mutually exclusive with a more sophisticated analysis of transitivity in trials with normal females, using female receptivity as a covariate or involving something more sophisticated, such as path analysis. Similarly, it would be important to know if the correlation between "male choice" and female productivity in fig. 3B is entirely mediated by female receptivity, or males mate with the more fecund female even if both females have the same receptivity.

I am sorry that I cannot be more positive. I do think this is a very interesting and extensive

set of results that will make a great contribution to the literature if presented in a balanced light.

Reviewer 1:

I have carefully read this new version of the MS, and I have to confess that, as a referee, I am impressed by the efforts made by the authors to address the reviewers's concerns. They not only modified the main text, discussed additional references, performed new (and much stronger) statistical analysis, but they also did complementary experiments. For instance, the new experiments (no choice experiment and decapitated females experiment) convincingly address the confounding effect of the female receptivity, and mutual mate choice. I consider that this manuscript is an important contribution (and a huge piece of work) which deserves publication in a high-impact factor journal.

My only minor recommendation would be to explain more carefully the non-trivial link between fitness and rationality. I believe that the excellent book chapter by Alex Kacelnick (2006. Meanings of rationality, pp. 87-105. Oxford: Oxford University Press) or the review by the MAD group (Fawcett, T. W., B. Fallenstein, A. D. Higginson, A. I. Houston, D. E. W. Mallpress, P. C. Trimmer, and J. M. McNamara. 2016. The evolution of decision rules in complex environments. Trends in Cognitive Sciences 18, 153-161) would be very helpful for the reader. I ask the authors to spend few lines to explain the concept of "ecological rationality" which is truly very different from the common sense of "rationality" in psychology for instance.

We have added additional discussion of the link between fitness and rationality, as well as the concept of ecological rationality to the introduction.

Lines 49 – 51

“Transitivity comes about because, the relative benefits of each choice should remain fixed, which leads to a linear rank order of options, and choosers should favour the highest ranking option in any scenario.”

Lines 56 – 63

“Recent theoretical work from an evolutionary perspective^{10,11} has also challenged the assumption that adaptive decision rules necessarily produce transitive choices. From this challenge, researchers have introduced the concept of “ecological rationality”¹¹, where choice rules evolve to maximize fitness in the environment to which a population has adapted. Because environments and the availability of alternative options can change rapidly, adaptive decision rules can produce non-transitive choices, though such choices are still rational as they maximize fitness.”

Reviewer 2:

I am happy with the changes that the authors have made, and look forward to seeing it published.

We thank the reviewer for their kind assessment.

Reviewer 3:

I have mixed feelings about this revised version of the ms.

On the one hand, I can see that the authors made a serious effort to address my comments, as well as those by the other reviewers. In particular, as I suggested, they changed the way they analyze transitivity and the correlation between female receptivity and the outcome of the mate choice assays. They added a correlation between time to first courtship in a one-to-one situation and the outcome of mate choice assays. And they performed, for a few pairs of female lines, a new experiment with male choice between headless females, thus eliminating female response. These last two results now show that male preference plays at least some role in the observed phenomenon. This is good, as no such evidence was present in the previous version.

On the other hand, I do not find that the authors have sufficiently addressed my criticism of their biased interpretation of the results as being entirely (or almost entirely) driven by male preference for females. The title, abstract, introduction and most of the result section are still written in terms of the results being unambiguously attributable to male mate choice, whereas in fact most of the assays reported (e.g. those described in l. 143 – 195) offer no possibility to distinguish the effect of male choice from those of female receptivity. Essentially, someone with as much bias for looking at female agency as these authors have for male agency could use these data to write a paper about the transitivity of female competitive prowess when competing for males. Although, as I said in the preceding paragraph, a couple of new results do support a role of male behavior, other results (notably the strong correlations between female receptivity in one-to-one trials and the outcome of "mate choice" trials) point to a (possibly greater) role of female behavior. This is especially the case with the new analysis, which shows such a correlation even for pkk23 males. I cannot avoid the impression that the authors are trying to sweep the inconvenient results under the carpet. E.g., despite their key importance for the interpretation of the mating trials, the correlations between female receptivity and the putative "male choice" is only described in supplementary material; in the Results section it is only alluded to in four (!) words (l. 192), and the potential role played by female behavior is not mentioned at all in the abstract.

The authors do express a more balanced view in the second paragraph of the discussion (l. 266ff), where they do admit that actually admit that what they found is that "female genotypes are sorted linearly with respect to the likelihood of mating", and while influence of both male and female behavior can be inferred, their relative importance remains unknown. This is good, but it means that there is currently a disconnect between this paragraph and the rest of the paper. The balanced attitude of this one paragraph should be extended to the entire paper. I believe an unbiased way of presenting and interpreting the results requires that the results of the 1 male/2 females mating trials be described in terms that are neutral with respect to male versus female agency. Finding the transitivity in these outcomes should then lead to the question of the relative contribution of male versus female behavior and the experiments that throw light on one versus another.

We have extensively revised our manuscript to represent the more balanced interpretations of our data that this reviewer advocates. We have changed the title, abstract, and every section of the main text to discuss the rationality of mate choice as a whole, rather than male choice specifically, and use more neutral language when discussing the results of the mate choice assays. We believe that we have removed the emphasis on male mate choice that was prevalent in the previous versions of this manuscript, and now discuss mate choice transitivity and the evidence for both male and female behaviour contributing to the rationality of mate choice. These extensive revisions can be seen in the highlighted changes throughout the manuscript (e.g. Lines 170 - 172, 332 - 336, 402 - 403). We believe that the manuscript is much more balanced in light of these revisions.

Furthermore, I reiterate my request for an analysis that reveals the degree to which the transitivity of the outcome of the putative "male choice" remains detectable after female receptivity is statistically controlled for. The authors argue in their response letter that they did the trials with headless females instead. However, this new experiment, while highly useful, only involves a subset of lines (understandably, as this is a large effort), and so it does not directly address the transitivity. I do not see why this new experiment should be mutually exclusive with a more sophisticated analysis of transitivity in trials with normal females, using female receptivity as a covariate or involving something more sophisticated, such as path analysis.

We have performed additional analyses to assess the observed mate choices after correcting for female receptivity. Though female receptivity is highly correlated with the mating biases of males in mate choice trials, the corrected data still shows transitive choices, which are similar to the choices in the uncorrected data.

Lines 745 - 755:

"To correct for the potential role of female receptivity in the mate choices of CS males, we calculated the residuals in the relationship between mating preferences for all pairwise comparisons between all lines, and the difference in female receptivity between the same lines. These residuals indicate the extent to which preferences for a specific female line were greater or lesser than expected based on female receptivity alone for each female line combination. From these residuals, we reassigned the mate choice of males (line 1 was preferred if mating bias for this line was greater than expected based on receptivity alone while line 2 was preferred if mating bias for this line was greater than expected). From this corrected preference matrix, we re-evaluated mate choice transitivity as described above."

Lines 262 - 303:

"We therefore attempted to statistically correct for female receptivity in the mating biases of CS males (see methods). When we accounted for variation explained by female receptivity, we found that the resulting corrected mate choices were significantly correlated with those from the uncorrected data ($r = 0.59$, $p = 0.0001$), and were still significantly transitive, although to a lesser degree than the uncorrected data ($t_{tri} = 0.88$, $p = 0.011$)."

Similarly, it would be important to know if the correlation between "male choice" and female productivity in fig. 3B is entirely mediated by female receptivity, or males mate with the more fecund female even if both females have the same receptivity.

We also analysed the relationship between mate choices and female productivity after correcting for female receptivity, and find that receptivity does appear to play a significant role in advantageous mate choices.

Lines 366 – 370

"We also evaluated the relationship between male mate choice after correction for female receptivity, and found that these adjusted preferences did not correlate with female offspring production ($r = 0.004$, $p = 0.98$). Therefore, once again, it appears that female behaviour plays an integral role in mediating the link between preferences and offspring production."

However, we also point out the limitations in interpretation from these additional analyses.

Lines 304 – 307

"While these analyses suggest that female receptivity plays a crucial role in producing the observed mate choices, by statistically correcting for the variance explained by female receptivity, we are potentially ignoring variance explained by non-behavioural traits that are correlated with female receptivity."

I am sorry that I cannot be more positive. I do think this is a very interesting and extensive set of results that will make a great contribution to the literature if presented in a balanced light.

We thank the reviewer for their comments and suggestions. We believe that our manuscript is much more balanced in light of the reviewers' recommendations.

REVIEWERS' COMMENTS:

Reviewer #3 (Remarks to the Author):

The authors have now addressed my concerns and I congratulate them on their neat work.
Tadeusz Kawecki